# Pupil-linked phasic arousal predicts a reduction of choice bias across species and decision domains

**Jan Willem de Gee[1,2,3,4]\*, Konstantinos Tsetsos[1], Lars Schwabe[5], Anne E Urai[1,2,6], David McCormick[7,8], Matthew J McGinley[3,4,8†]\*, Tobias H Donner[1,2,9†]\***

[1]Department of Neurophysiology and Pathophysiology, University Medical Center Hamburg-Eppendorf, Hamburg, Germany; [2]Department of Psychology, University of Amsterdam, Amsterdam, Netherlands; [3]Department of Neuroscience, Baylor College of Medicine, Houston, United States; [4]Jan and Dan Duncan Neurological Research Institute, Texas Children's Hospital, Houston, United States; [5]Department of Cognitive Psychology, Institute of Psychology, Universität Hamburg, Hamburg, Germany; [6]Cold Spring Harbor Laboratory, Cold Spring Harbor, United States; [7]Institute of Neuroscience, University of Oregon, Eugene, United States; [8]Department of Neuroscience, Yale University, New Haven, United States; [9]Amsterdam Brain and Cognition, University of Amsterdam, Amsterdam, Netherlands

**\*For correspondence:**
jwdegee@gmail.com (JWG);
Matthew.McGinley@bcm.edu
(MJMG);
t.donner@uke.de (THD)

[†]These authors contributed equally to this work

**Abstract** Decisions are often made by accumulating ambiguous evidence over time. The brain's arousal systems are activated during such decisions. In previous work in humans, we found that evoked responses of arousal systems during decisions are reported by rapid dilations of the pupil and track a suppression of biases in the accumulation of decision-relevant evidence (de Gee et al., 2017). Here, we show that this arousal-related suppression in decision bias acts on both conservative and liberal biases, and generalizes from humans to mice, and from perceptual to memory-based decisions. In challenging sound-detection tasks, the impact of spontaneous or experimentally induced choice biases was reduced under high phasic arousal. Similar bias suppression occurred when evidence was drawn from memory. All of these behavioral effects were explained by reduced evidence accumulation biases. Our results point to a general principle of interplay between phasic arousal and decision-making.

## Introduction

The global arousal state of the brain changes from moment to moment (**Aston-Jones and Cohen, 2005**; **McGinley et al., 2015b**). These global state changes are controlled in large part by modulatory neurotransmitters that are released from subcortical nuclei such as the noradrenergic locus coeruleus (LC) and the cholinergic basal forebrain. Release of these neuromodulators can profoundly change the operating mode of target cortical circuits (**Aston-Jones and Cohen, 2005**; **Froemke, 2015**; **Harris and Thiele, 2011**; **Lee and Dan, 2012**; **Pfeffer et al., 2018**). These same arousal systems are phasically recruited during elementary decisions, in relation to computational variables such as uncertainty about making the correct choice and surprise about decision outcome (**Aston-Jones and Cohen, 2005**; **Bouret and Sara, 2005**; **Colizoli et al., 2018**; **Dayan and Yu, 2006**; **Krishnamurthy et al., 2017**; **Lak et al., 2017**; **Nassar et al., 2012**; **Parikh et al., 2007**; **Urai et al., 2017**).

Most decisions – including judgments about weak sensory signals in noise – are based on a protracted accumulation of decision-relevant evidence (*Shadlen and Kiani, 2013*) implemented in a distributed network of brain regions (*Pinto et al., 2019*). In perceptual decisions, noise-corrupted decision evidence is encoded in sensory cortex, and downstream regions of association and motor cortices accumulate the fluctuating sensory response over time into a decision variable that forms the basis of behavioral choice (*Bogacz et al., 2006*; *Shadlen and Kiani, 2013*; *Siegel et al., 2011*; *Wang, 2008*). All of these brain regions are impacted by the brain's arousal systems. Thus, phasic arousal might shape the encoding of the momentary evidence, the accumulation of this evidence into a decision variable, and/or the implementation of the motor act.

We previously combined fMRI, pupillometry and behavioral modeling in humans to illuminate the interaction between phasic arousal and perceptual evidence accumulation (*de Gee et al., 2017*). We found that rapid pupil dilations during perceptual decisions report evoked responses in specific neuromodulatory (brainstem) nuclei controlling arousal, including the noradrenergic LC. We also showed that those same pupil responses track a suppression of pre-existing biases in the accumulation of perceptual evidence. Specifically, in perceptual detection tasks, spontaneously emerging 'conservative' biases (towards reporting the absence of a target signal) were reduced under large phasic arousal. Thus, it remains an open question whether phasic arousal promotes liberal decision-making, or suppresses biases in either direction (conservative and liberal).

It is also unclear whether the impact of arousal generalizes to decisions that are based on non-sensory evidence, such as memory-based decisions. Elementary perceptual choice tasks are an established laboratory approach to studying decision-making mechanisms, but many important real-life decisions (e.g. which stock to buy) are also based on information gathered from memory. Recent advances indicate that such decisions may entail the accumulation of decision-relevant evidence drawn from memory (*Shadlen and Shohamy, 2016*).

Finally, it is also unknown whether the impact of arousal on decision-making generalizes from humans to rodents. This is important because rodents are increasingly utilized as experimental models for studying decision-making mechanisms (*Carandini and Churchland, 2013*; *Najafi and Churchland, 2018*). Indeed, rodents (rats) can accumulate perceptual evidence in a similar fashion to that in humans (*Brunton et al., 2013*), their arousal systems are organized homologously with those of humans (*Amaral and Sinnamon, 1977*; *Berridge and Waterhouse, 2003*), and pupil dilation reports arousal also in rodents (*McGinley et al., 2015b*; *Reimer et al., 2014*; *Vinck et al., 2015*). But the interplay between phasic arousal and decision-making in rodents remains unknown.

Here, we address the issues pertaining to the interplay between phasic arousal and decision-making outlined above. Specifically, we asked (i) whether the phasic arousal-linked suppression of decision biases generalizes from humans to rodents; (ii) whether phasic arousal predicts liberal decision-making, or a suppression of both liberal and conservative biases; and (iii) whether the interaction between phasic arousal and decision biases generalizes from perceptual to memory-based decisions. We addressed these questions by combining pupillometry and computational model-based analyses of behavior, in both humans and mice (*Badre et al., 2015*), and by studying human decision-making in several contexts.

## Results

We measured pupil-indexed phasic arousal while humans and mice performed the same auditory go/no-go detection task. To test for generality across domains of decision-making, humans performed a forced choice (yes/no) detection task that was based on the same auditory evidence under systematic manipulations of target probabilities, as well as a memory-based decision task.

### In humans and mice, phasic arousal tracks a reduction of choice bias

We first trained mice (N = 5) and humans (N = 20) to detect a near-threshold auditory signal (*Figure 1A,B*; 'Materials and methods'). Each trial was a distinct auditory noise stimulus of 1 s duration. A weak signal tone was added to the last trial in a mini block of two-to-seven consecutive trials (*Figure 1A*). The number of trials, and thus the signal position in the sequence, was drawn randomly

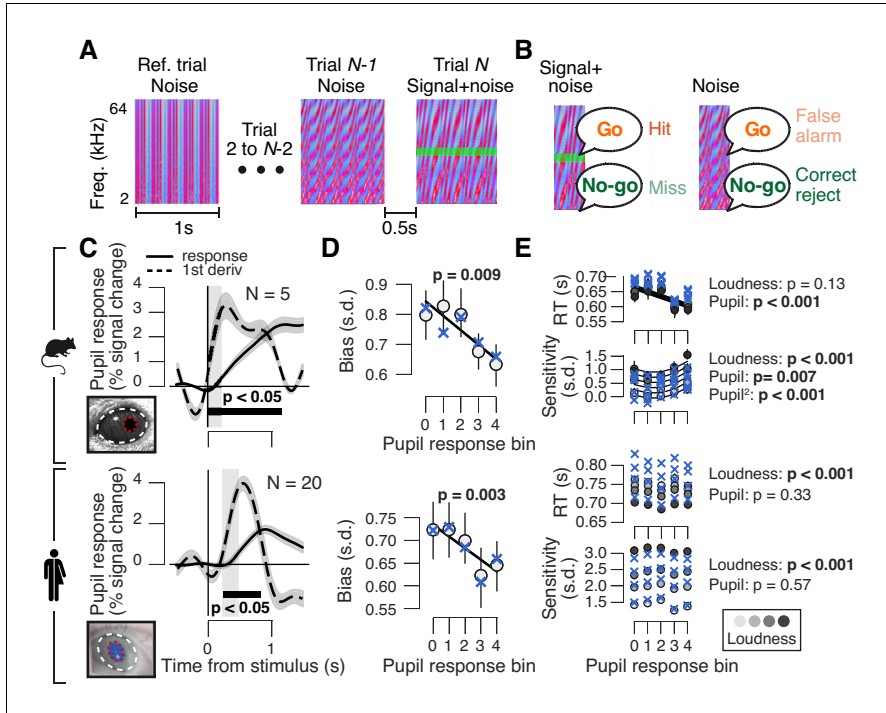

**Figure 1.** High phasic arousal is associated with reduced perceptual choice bias. (**A**) Auditory go/no-go detection task. Schematic sequence of discrete trials in a mini block (see 'Materials and methods'). Subjects responded to a weak signal (stable pure tone) in fluctuating noise and withheld a response for noise-only trials. (**B**) Combination of stimulus (signal+noise vs. noise) and choice (go vs. no-go) yielded the four categories of signal detection theory. (**C**) Change in pupil diameter (solid line) and its temporal derivative (dashed) in mice (top) and humans (bottom). Gray window, interval for extracting task-evoked pupil responses (see 'Materials and methods'); black bar, significant pupil derivative (p<0.05, cluster-corrected one-sample t-test). (**D**) Relationship between pupil response (equal size bins, see 'Materials and Methods') and choice bias in mice (top) and humans (bottom). We assumed that subjects set a single decision criterion (see also *Figure 1—figure supplement 1B,H*). (**E**) As panel (D), but for mean reaction time (RT) and sensitivity (d'). For panels (C–E): group average (N = 5; N = 20); shading or error bars, s.e.m. Solid lines, linear or quadratic fits to binned data (linear fits are shown where first-order fit was superior to constant fit, quadratic fits are shown where second-order fit was superior to first-order fit). Blue 'X's, predictions from the best fitting drift diffusion model (see *Figure 4* and associated text); p-values, mixed linear modeling (predictors are 'loudness' = signal loudness and 'pupil' = pupil bin).

The online version of this article includes the following figure supplement(s) for figure 1:

**Figure supplement 1.** Additional analyses of data from auditory go/no-go detection task.

---

with signal probability that decreased across the mini block (*Figure 1—figure supplement 1A*, left). Because stable signals were embedded in fluctuating noise, detection performance could be maximized by accumulating the sensory evidence over time, within each trial. To indicate a yes choice, mice licked for sugar water reward and human subjects pressed a button. Reaction times (RTs) decreased and perceptual sensitivity (d', from signal detection theory; see 'Materials and methods') increased with tone loudness (*Figure 1—figure supplement 1D,E*). We quantified phasic arousal as the rising slope of the pupil, measured immediately after trial onset (see 'Materials and methods'). We chose this measure (i) for its temporal precision in tracking arousal during fast-paced tasks (*Figure 1C*), (ii) to eliminate contamination by movements, and (iii) to track noradrenergic activity most specifically (*Reimer et al., 2016*), which may play a specific role in decision-making (*Aston-Jones and Cohen, 2005*; *Dayan and Yu, 2006*). Thus, we use the pupil response derivative as a proxy for the amplitude of the phasic response of central arousal systems. In what follows, we refer to this as the 'pupil response' for simplicity.

Pupil responses occurred as early as 40 ms after trial onset in mice (*Figure 1C*, top), and from 240 ms after trial onset in humans (*Figure 1C*, bottom). Because trial timing was predictable,

subjects were able to anticipate the sound starts, and to align their arousal response to trial onset. The shorter pupil response latencies in mice compared to humans might be due to species differences in impulsivity, timing estimation and/or physical properties (their smaller eye and brain size).

Pupil dilations occurred on all trials, whether or not there was a behavioral response (*Figure 1—figure supplement 1F*), as also observed previously (*Lee and Margolis, 2016*; *Schriver et al., 2020*). However, as in our earlier work (*de Gee et al., 2017*), we found a relationship between the early, task-evoked pupil response and choice bias. Both mice and humans had an overall conservative choice bias, often failing to report the signal tones (*Figure 1D*; bias computed after collapsing across signal loudness levels; see also *Figure 1—figure supplement 1B,H* and 'Materials and methods'). In both species, this conservative bias was significantly reduced in trials characterized by large pupil responses (*Figure 1D*). Mice were faster and more sensitive in trials that were characterized by large pupil responses, but this relationship did not generalize to humans (*Figure 1E*).

## Correlation between pupil response and choice bias is not due to motor factors

One concern in the go/no-go task is that the bias suppression associated with pupil dilation was related to the motor response for go choices. The central input to the pupil contains a transient around the motor response (*de Gee et al., 2014*; *de Gee et al., 2017*; *Hupé et al., 2009*; *Murphy et al., 2016*). Such transient activity at lick or button-press responses would contribute to pupil responses in trials with go-choices, but not in trials with no-go choices. This might produce the observed correlation between mean pupil response (more motor transients) and more liberal choice bias (more go-choices). We here minimized contamination by the transient motor-related component by focusing on the very early component of pupil response (see 'Materials and methods').

However, this approach did not correct for any asymmetry between go and no-go trials in terms of motor preparatory activity, which commonly ramps up during perceptual decision formation seconds before response execution (*Donner et al., 2009*; *Shadlen and Kiani, 2013*). Also the central input to the pupil during decision formation contains a sustained component, which might, at least in part, be related to motor preparatory activity (*de Gee et al., 2014*; *de Gee et al., 2017*; *Murphy et al., 2016*). To address this concern further, we re-analyzed results from the forced-choice (yes/no) version of the task (*Figure 2A*; 'Materials and methods'), which were published previously (*de Gee et al., 2017*) and which partly stemmed from the same participants as those that produced the go/no-go data. In yes/no detection tasks, both motor responses and associated motor preparatory activity are balanced across yes and no choices (*Donner et al., 2009*). We found that pupil responses in the go/no-go and yes/no tasks were correlated across participants, for both yes and no choices (*Figure 2—figure supplement 1A*). This result indicates that pupil response in both tasks primarily reflected decision processing rather than motor preparation or execution. Furthermore, we observed an arousal-linked suppression of perceptual choice bias (*Figure 2E*, top). Taken together, the results indicate that the pupil-linked reduction of choice bias (*Figure 1* and *2*) is not due to motor factors.

## Phasic arousal tracks a reduction of both conservative and liberal perceptual choice biases

All mice and humans in the above go/no-go protocol, and the majority of human subjects in the above yes/no task, exhibited a conservative bias, or tendency to choose 'no', which was reduced in trials with large pupil response (*Figure 1D* and *Figure 2E*, top). We thus wondered whether phasic arousal predicts liberal decision-making or rather a suppression of any pre-existing bias, be it liberal or conservative. To address this, we asked a new group of human subjects (N = 15) to perform the same auditory yes/no (forced-choice) detection task, but with a different probability of signal occurrence in blocks. In 'rare' blocks, the signal occurred in 30% of trials, whereas in 'frequent' blocks, the signal occurred in 70% of trials (*Figure 2B*; see 'Material and methods'). As expected (*Green and Swets, 1966*), subjects developed a conservative bias in the rare signal condition and a liberal bias in the frequent signal condition (*Figure 2C*). Importantly, pupil response predicted a change in choice biases towards neutral for both block types (performed within the same experimental session) (*Figure 2E*). As in the go/no-go task, pupil responses were not consistently associated with RT or

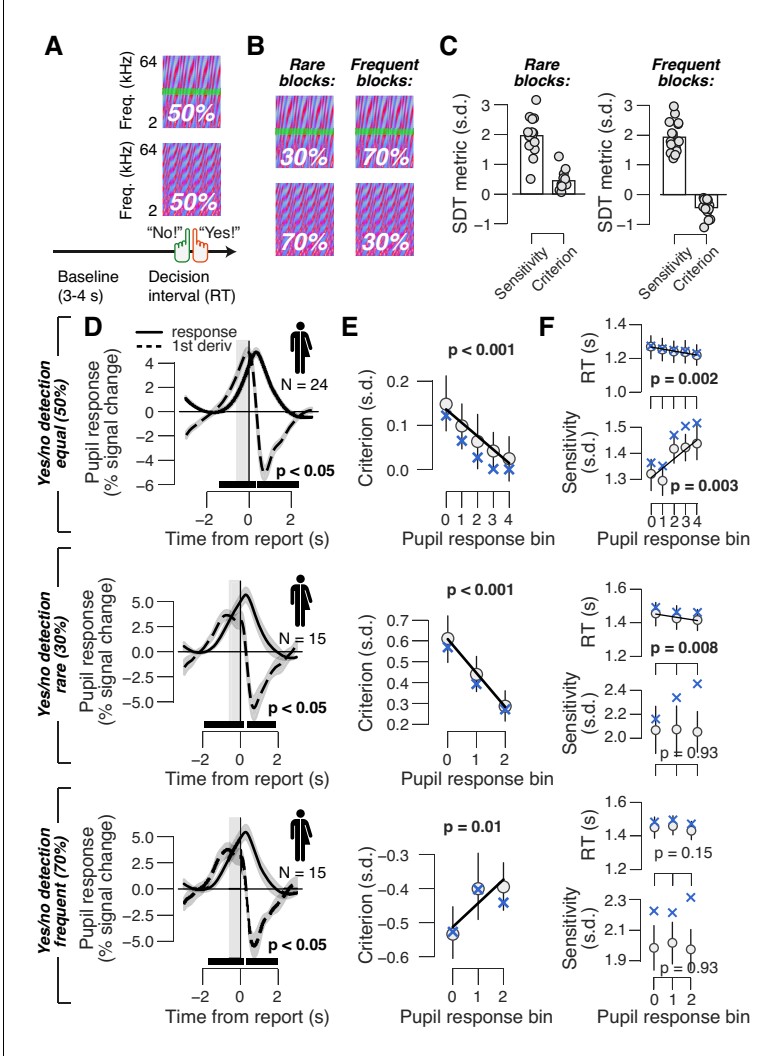

**Figure 2.** Phasic arousal reduces both conservative and liberal choice biases. (**A**) Auditory yes/no (forced choice) tone-in-noise detection task. Schematic sequence of events during a trial. Subjects reported the presence or absence of a faint signal (pure tone; green band) embedded in noise (see 'Materials and methods'). (**B**) A separate batch of subjects performed the same task, but the signal now occurred in 30% of trials ('rare' condition) or in 70% trials ('frequent' condition). (**C**) Overall sensitivity and bias in rare and frequent conditions. (**D**) Task-evoked pupil response (solid line) and its derivative (dashed) in the equal (top), rare (middle) and frequent (bottom) signal occurrence conditions. Gray window, interval for extracting task-evoked pupil response measures; black bar, significant pupil derivative (p<0.05, cluster-corrected one-sample t-test). (**E**) Relationship between pupil response and choice bias in the equal (top), rare (middle) and frequent (bottom) signal occurrence conditions. For the biased signal occurrence conditions, we used three pupil-defined bins because there were fewer trials per individual (less than 500) than in the previous data sets (more than 500; see 'Materials and methods'). (**F**) As panel (**E**), but for mean RT and perceptual sensitivity. (**D–F**) Group average (N = 24; N = 15; N = 15); shading or error bars = s.e.m. Solid lines show linear or quadratic fits to binned data (linear fits are shown where first-order fit was superior to constant fit; quadratic fits are shown where second-order fit was superior to first-order fit). Blue 'X's, predictions from the 'full' drift diffusion model (see *Figure 4* and associated text); p-values, mixed linear modeling.

The online version of this article includes the following figure supplement(s) for figure 2:

**Figure supplement 1.** Additional analyses of data from auditory yes/no detection tasks.

sensitivity (*Figure 2F*). In sum, phasic arousal predicts a reduction of choice biases irrespective of direction (conservative or liberal).

## Phasic arousal tracks a reduction of biases in memory-based decisions

We next assessed whether the arousal-related bias suppression identified for perceptual decisions generalizes to memory-based decisions. We characterized the interaction between pupil-linked phasic arousal and choice behavior in a yes/no picture recognition task (*Figure 3A*; see 'Materials and methods'; *Bergt et al., 2018*). Subjects (N = 54) were instructed to memorize 150 pictures (intentional encoding) and to evaluate how emotional each picture was on a 4-point scale from 0 ('neutral') to 3 ('very negative'). Twenty-four hours after encoding, subjects saw all pictures that were presented on the first day and an equal number of novel pictures in randomized order, and indicated for each item whether it had been presented before ('yes – old') or not ('no – new'). Subjects' overall biases (irrespective of pupil response) varied from strongly liberal to strongly conservative across the 54 individuals (*Figure 3C,* x-axis and colors). Therefore, this data set also afforded an across-subjects test of the direction-dependence (conservative vs liberal) of the pupil-linked arousal effect, which was complementary to the within-subject test reported in the previous section.

Indeed, we observed a robust relationship between subjects' overall choice bias, and the pupil-linked shift in that bias: subjects with the strongest overall biases, whether liberal or conservative, exhibited the strongest pupil-linked shift towards neutral bias (*Figure 3C*). Correspondingly, the absolute value of the of the bias, measuring the magnitude of bias irrespective of sign, was significantly reduced towards 0 in high-pupil bins (*Figure 4D*). In this experiment, pupil responses were positively correlated with RT, but not with sensitivity (*Figure 4E*). In summary, phasic arousal is also associated with a suppression of both liberal and conservative biases in memory-based decisions.

## Pupil-linked bias reduction is associated with changes in the evidence accumulation process

To gain deeper insight into the interaction of phasic pupil-linked arousal and the dynamics of the decision process, we fitted bounded accumulation models of decision-making to the behavioral measurements (choices and RTs; *Figure 4A*). We fitted several variants of a popular version of such models, the drift diffusion model (*Bogacz et al., 2006*; *Brody and Hanks, 2016*; *Gold and Shadlen, 2007*; *Ratcliff and McKoon, 2008*). The diffusion model describes the perfect accumulation of noisy sensory evidence in a single decision variable that drifts to one of two decision bounds (*Figure 4A*). For all yes/no data sets, we quantified the effects of pupil-linked arousal on the following model

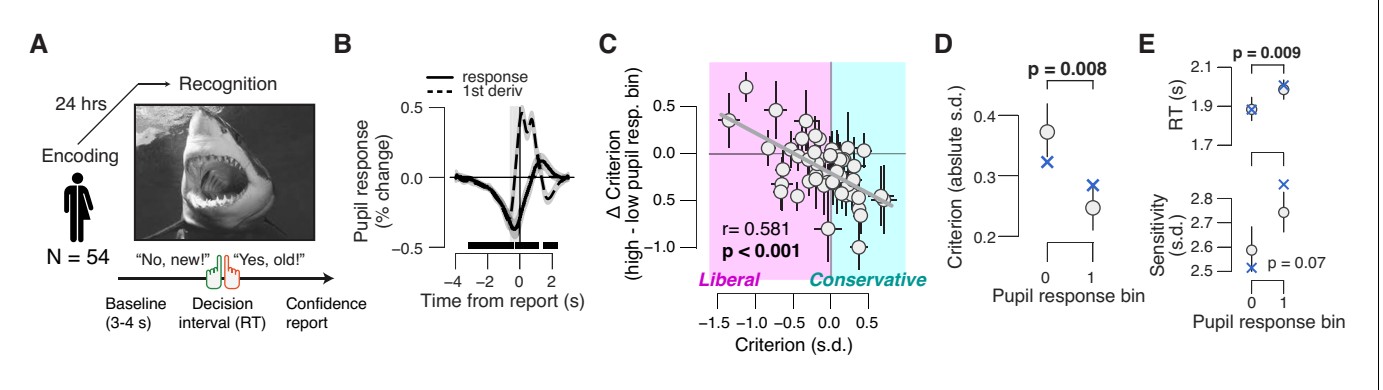

**Figure 3.** Phasic arousal tracks a reduction of memory biases. (**A**) A yes/no (forced choice) picture recognition task. Schematic sequence of events during a trial. Subjects judged whether they had seen pictures 24 h previously during an encoding task (see 'Materials and methods'). (**B**) Task-evoked pupil response (solid line) and response derivative (dashed line). The gray window shows the interval for extracting task-evoked pupil response measures; the black bars indicate a significant pupil derivative (p<0.05, cluster-corrected one-sample t-test). (**C**) Individual pupil-linked shift in choice bias, plotted against individual's overall choice bias. Data points are individual subjects. Correlation was assessed statistically by Pearson's correlation coefficient. Error bars represent 60% confidence intervals (bootstrap). (**D**) Relationship between magnitude of choice bias (absolute value) and task-evoked pupil response. Difference was assessed using paired-samples t-test. (**E**) As panel (D), but for mean RT and perceptual sensitivity. In (B, D,E), data are shown as group averages (N = 54); shading or error bars represent the s.e.m. Blue 'X's show predictions from the 'full' drift diffusion model (see *Figure 4* and associated text).

The online version of this article includes the following figure supplement(s) for figure 3:

**Figure supplement 1.** Additional analyses of data from yes/no picture recognition task.

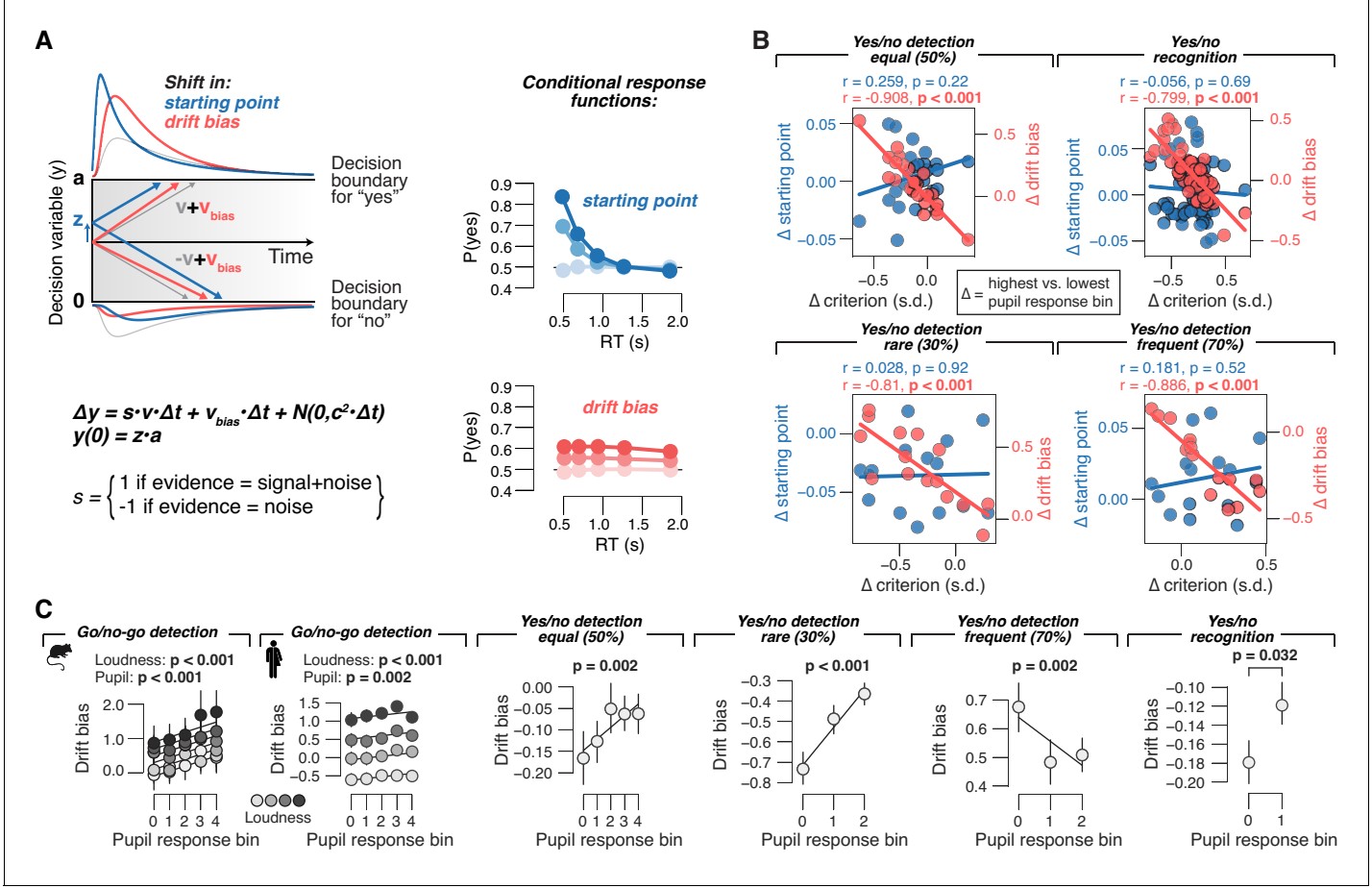

**Figure 4.** Phasic arousal tracks changes in evidence accumulation bias. (A) Schematic of drift diffusion model accounting for choices, and their associated RTs. In the equation, *v* is the drift rate. Red and blue curves on the left show expected RT distributions under shifts in either the 'starting point' ($z$; blue) or 'drift bias' model ($v_{bias}$; red). Conditional response functions (right) were generated by dividing synthetic RT distributions (see 'Materials and methods') into five quantiles and computing the fraction of yes choices (upper boundary choices) in each quantile. (B) Individual pupil-linked shift in starting point (blue) or drift bias (red), plotted against individual's pupil-linked shift in choice bias. Data points are individual subjects. Pearson's correlation coefficients are shown. (C) Relationship between drift bias and task-evoked pupil response, separately for each data set (left to right). In the picture recognition data set, drift bias estimates were sign-flipped for overall liberal subjects. Solid lines, linear or quadratic fits to binned data (linear fits are shown where first-order fit was superior to constant fit; quadratic fits are shown where second-order fit was superior to first-order fit; p-values were calculated using mixed linear modeling (predictors 'loudness' [signal loudness] and 'pupil' [pupil bin]), or paired-sample t-tests for the picture recognition data set. Group average (N = 5, N = 20, N = 24, N = 15, N = 15 and N = 54, respectively); error bars represent s.e.m. The online version of this article includes the following figure supplement(s) for figure 4:

**Figure supplement 1.** Drift diffusion model fit and comparisons.
**Figure supplement 2.** Remaining drift diffusion model parameters as function of pupil response bin.
**Figure supplement 3.** Empirical conditional response functions in the four yes/no data sets (left to right), separately for highest and lowest pupil bin.

parameters (see 'Materials and methods'): 1) the starting point of the decision, 2) the mean drift rate, 3) an evidence-independent bias in the drift (henceforth called 'drift bias'), 4) boundary separation (controlling speed-accuracy tradeoff), and 5) non-decision time (the speed of pre-decisional evidence encoding and post-decisional translation of choice into motor response). The diffusion model accounted well for the overall behavior in each task (*Figure 4—figure supplement 1A*), with accurate predictions of RT, sensitivity and bias (blue 'X' markers in *Figures 1–3*), and the expected increase of drift rate with signal loudness (*Figure 4—figure supplement 2C*).

Within the diffusion model, a bias might be due to a shift in the starting point or to a bias in the drift (alone or in combination with another model parameter, see below; *Figure 4A*). Starting point and drift bias have dissociable effects on the shape of the RT distribution, which are evident in the 'conditional response functions' (*Figure 4A*, right; *White and Poldrack, 2014*). The conditional

response functions of all yes/no data sets exhibited pupil-linked shifts across the full range of RTs (*Figure 4—figure supplement 3*), a pattern more consistent with a shift in drift bias. More importantly, the individual pupil-linked changes in drift bias, but not in starting point, strongly correlated with the corresponding changes in the overt behavioral biases (*Figure 4B*).

For the forced choice (yes/no) data sets, we allowed all of the model parameters listed above to vary as a function of pupil response (see 'Materials and methods'). For the go/no-go task, the model was not fully constrained because of the absence of RTs for no choices. This precluded us from fitting both bias parameters (starting point or drift bias) as a function pupil response bin. Because model comparisons favored pupil-dependent variation of drift bias over pupil-dependent starting point (*Figure 4—figure supplement 1B*), we proceeded with the former version, estimating a single starting point value irrespective of pupil bin.

We also formally compared the fit provided by the 'full model' described before to two alternative models: (i) the 'starting point model', with drift bias fixed across pupil bins and starting point varying with pupil bins (otherwise identical to the full model); and (ii) the 'drift bias model', with starting point fixed across pupil bins and drift bias varying with pupil bins (otherwise identical to the full model). The starting point model provided a worse fit than the full model in three out of four data sets (*Figure 4—figure supplement 1B*). The drift bias model provided a better fit than the full model (and than the starting point model) in all four data sets (*Figure 4—figure supplement 1B*).

In line with the results described above, pupil responses predicted linear shifts in drift bias in all data sets (*Figure 4C*), but not (consistently) in starting point (for yes/no data; *Figure 4—figure supplement 2A*). In the go/no-go data sets from both species, starting point was biased towards no-go (*Figure 4—figure supplement 2A*). Overcoming this conservative bias set by starting point required shifting the drift bias towards the yes bound, which occurred in trials with rapid pupil dilation (*Figure 4C*). The linear relationship between pupil responses and drift bias (*Figure 4C*) accurately tracked the pupil-dependent reduction in overt conservative choice bias (blue 'X' markers in *Figures 1D* and *2D*). In summary, the pupil-linked behavioral effects in all of the data sets analyzed here were consistent with a phasic arousal-dependent, selective reduction in the bias of the evidence accumulation process that underlies the decisions assessed here.

Some accounts hold that arousal specifically enhances the representation of task-relevant variables (*Aston-Jones and Cohen, 2005*; *Mather et al., 2016*). In the tasks used here, this scenario predicts increased sensitivity and/or reduced RT under high arousal. The pupil-linked effects in d' or RT in several of the data sets analyzed here (*Figures 1–3*) are inconsistent with this prediction. Correspondingly, pupil dilation also exhibited no consistent (across data sets) association with changes of drift rate (measure of sensitivity), and no association with boundary separation (measure of response caution) or non-decision time (*Figure 4—figure supplement 2B–D*).

Our results imply that the build-up (drift) of the decision variable varies dynamically across trials, irrespective of the sensory evidence but as a function of phasic arousal. The resulting variability in the drift would appear to be random when ignoring phasic arousal. Such random variability in drift is captured by the drift rate variability parameter of the diffusion model (*Bogacz et al., 2006*; *Ratcliff and McKoon, 2008*). We solidified this intuition by simulating RT distributions from two conditions that differed with regard to drift bias (*Figure 5*; see 'Materials and methods'), just like trials with high vs low phasic arousal responses in our experiments. When we allowed drift bias to vary with condition (as we did in our pupil-dependent model fits above), we were actually able to recover the true drift rate variability of the generative process (*Figure 5A*). But when

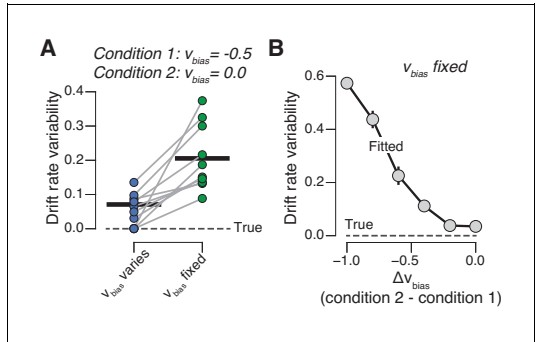

**Figure 5.** Untracked changes in drift bias account for drift rate variability. (**A**) Recovered drift rate variability for models with fixed or varying drift bias. The model was fit to simulated RT distributions (N = 10) from two conditions that differed according to drift bias (see 'Materials and methods'). The dashed line represents true (simulated) drift rate variability. (**B**) Recovered drift rate variability for a model with fixed drift bias, for conditions that differed according to varying levels of drift bias (see 'Materials and methods').

we forced the drift bias to be the same for both conditions, drift rate variability was overestimated (*Figure 5A*). This additional ('apparent') component of drift rate variability increased monotonically with the disparity of drift biases between both conditions (*Figure 5B*).

Thus, a significant fraction of choice variability does not originate from noise within the evidence accumulation machinery itself, but rather from dynamic, arousal-linked variations in evidence accumulation bias. This insight should be incorporated into recent accounts that attribute trial-by-trial choice variability to evidence accumulation noise rather than to biases, under the assumption of constant biases (*Drugowitsch et al., 2016*). Note that we are agnostic about the source of trial-by-trial variations in phasic arousal, which was not under experimental control in the present study (but see *Colizoli et al., 2018*; *Nassar et al., 2012*; *Urai et al., 2017*).

## Changes in urgency are unlikely to mediate the pupil-linked reduction of choice bias

In the previous section, we assumed that the decision bounds remained constant during decision formation (*Ratcliff and McKoon, 2008*). In reality, decision bounds may often collapse over time (*Churchland et al., 2008*; *Murphy et al., 2016*; *Urai et al., 2019*). This implements a form of 'decision urgency' signal (*Cisek et al., 2009*), as less total evidence is required to cross threshold after a long interval of decision processing than at the start of the decision process, preventing decision deadlock. Pupil-linked phasic arousal might also control the dynamics of decision urgency (*Murphy et al., 2016*). In a final model approach, we assessed whether a pupil-dependent variation of collapsing bounds (urgency), when combined with pupil-independent starting point or drift bias, could account for the pupil-dependent changes in behavioral bias reported in *Figures 1–3*. Converging results from model simulations and fits deem this scenario unlikely (*Figure 6*, *Figure 6—figure supplement 1*).

Simulations of models equipped with different biasing mechanisms (varying starting point or drift bias without urgency; or with fixed starting point/drift bias combined with varying urgency) showed that only changes in drift bias were qualitatively in line with the empirical data, in that they changed choice biases without producing large concomitant changes in accuracy or RT (data not shown).

We then fitted the yes/no data sets to three alternative models, each with the same number of parameters (*Figure 6A* and 'Materials and methods'): (i) the starting point model, with starting point varying with pupil bin and one overall drift bias (no urgency); (ii) the drift bias model, with one overall starting point bias, and drift bias varying with pupil bin (no urgency); and (iii) the urgency model, with one overall starting point bias, and no drift bias (urgency varying with pupil bin). (A fixed drift bias combined with varying urgency was not considered here because our simulations showed that this could not produce any biases of the observed magnitudes.) In each model, we fitted boundary separation, drift rate and non-decision time irrespective of pupil bin. For formal model comparison of these models, see *Figure 6—figure supplement 1A*). Finally, we simulated data on the basis of the fitted parameters of the above starting point, drift bias and urgency models, and assessed the corresponding variations in overt choice biases as a function of pupil response bin. The model with collapsing bounds produced only negligible pupil-linked variations in choice bias (*Figure 6B*), contrary to what we observed in all data sets (*Figures 1–3*). We then compared the differences between the choice biases predicted by each model and the empirically measured choice biases, referred to as 'residuals' for simplicity. In each data set, the residuals of the urgency model were significantly larger than those of both the starting point and drift bias models (*Figure 6C*). In each data set, the residuals of the drift bias model were smallest (*Figure 6C*), indicating that this model best captured the pupil-dependent variation in overt choice bias. Taken together, all of our modeling efforts converged on the conclusion that changes in drift bias, but in neither starting point nor decision urgency (combined with a pre-existing starting point bias), implement the pupil-linked shifts in choice bias.

## Suppression of bias by phasic arousal is distinct from ongoing slow state fluctuations

One concern is that the bias suppression related to phasic arousal reported here might be 'inherited' through a previously observed negative correlation between phasic pupil responses and pre-trial baseline pupil diameter (*de Gee et al., 2014*; *Gilzenrat et al., 2010*; *Mridha et al., 2019*). This is probably due to a combination of a true negative correlation between baseline and evoked arousal,

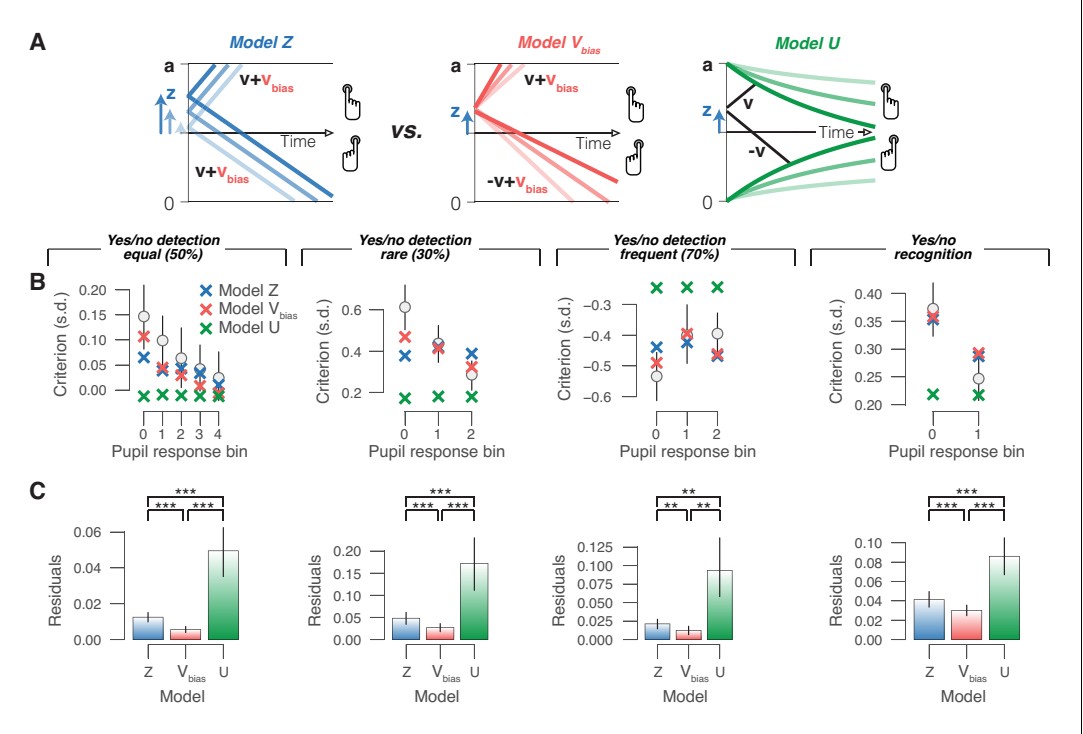

**Figure 6.** Phasic arousal reduces overt choice bias by reducing a bias in evidence accumulation. (A) Schematic of three alternative models. (B) Relationship between choice bias and task-evoked pupil response, separately for each yes/no data set (left to right). 'X' symbols are predictions from the three alternative models in panel (A) (see 'Materials and methods'). (C) Residuals between empirical data and model predictions (see 'Materials and methods'). Statistical difference was assessed by paired-sample t-tests. In panels (B) and (C), data are group averages (N = 24, N = 15, N = 15 and N = 54); error bars are s.e.m.

The online version of this article includes the following figure supplement(s) for figure 6:

**Figure supplement 1.** Comparison between fits of models with or without varying urgency.

and a general tendency of pupil size to revert to the mean (*Mridha et al., 2019*). Pre-trial baseline pupil size might vary from trial to trial because it tracks slow (ongoing) arousal fluctuations and/or is contaminated by pre-trial spill-over effects (e.g., uncertainty [high/low], response [yes/no], and pupil response magnitude).

A dependence on baseline pupil could not account for the results reported here, for four reasons. First, there was a non-monotonic association between baseline pupil diameter and decision bias in mice (*McGinley et al., 2015a*), in contrast to the monotonic pattern that we observed here for phasic arousal in the same data set (*Figure 1D*). Second, pupil responses exhibited a negligible (go/no-go data sets) or weak (yes/no data sets) correlation with the preceding baseline diameter (*Figure 1—figure supplement 1H*, *Figure 2—figure supplement 1D* and *Figure 3—figure supplement 1C*); in all cases, these correlations are substantially smaller than those for conventional baseline-corrected pupil response amplitudes (*de Gee et al., 2014*; *Gilzenrat et al., 2010*). Third, in three out of four data sets with a weak anticorrelation between pupil response and baseline pupil size, there was no consistent linear association between baseline pupil diameter and choice bias (yes/no equal, p=0.002; yes/no rare, p=0.265; yes/no frequent, p=0.157; yes/no recognition, p=0.592; using the same mixed-linear modeling approach as in *Figures 1–3*). In summary, the reduction in choice bias linked to pupil responses reported here reflects genuine correlates of phasic arousal, and not of the slowly varying, baseline arousal state.

## Discussion

Arousal is traditionally thought to upregulate the efficiency of information processing globally (e.g., the quality of evidence encoding or the efficiency of accumulation [*Aston-Jones and Cohen, 2005*;

*Mather et al., 2016*; *McGinley et al., 2015b*]). However, recent work indicates that phasic arousal signals might have distinct effects, such as reducing the impact of prior expectations and biases on decision formation (*de Gee et al., 2014*; *de Gee et al., 2017*; *Krishnamurthy et al., 2017*; *Nassar et al., 2012*; *Urai et al., 2017*). Our results are in consistent with the idea that phasic arousal suppresses biases in the accumulation of evidence leading up to a choice, a function that generalizes across species (humans and mice) and domains of decision-making (from perceptual to memory-based).

We observed pupil-linked bias-reduction in human and mouse choice behavior during analogous auditory tone-detection tasks. Task-evoked pupil responses occurred early during decision formation, even in trials without a motor response, and tracked a suppression of conservative choice bias. Behavioral modeling indicated that the bias reduction was due to a selective interaction with the accumulation of information from the noisy sensory evidence. We further showed that phasic arousal tracks a reduction of accumulation biases, whether conservative or liberal, as seen in the conditions with different stimulus presentation statistics. Finally, we showed, under the assumption that mnemonic decisions are based on samples from memory, that pupil-linked suppression of evidence accumulation bias also occurs for memory-based decisions. We conclude that the ongoing deliberation, culminating in a choice (*Shadlen and Kiani, 2013*), interacts in a canonical way with transient boosts in the global arousal state of the brain: more phasic arousal tracks reduced evidence accumulation bias.

We here used pupil responses as a peripheral readout of changes in cortical arousal state (*Joshi and Gold, 2020*; *Larsen and Waters, 2018*; *McGinley et al., 2015b*). Indeed, recent work has shown that pupil diameter closely tracks several established measures of cortical arousal state (*Larsen and Waters, 2018*; *McGinley et al., 2015a*; *McGinley et al., 2015b*; *Reimer et al., 2014*; *Vinck et al., 2015*). Changes in pupil diameter have been associated with locus coeruleus (LC) activity in humans (*de Gee et al., 2017*; *Murphy et al., 2014*), monkeys (*Joshi et al., 2016*; *Varazzani et al., 2015*), and mice (*Breton-Provencher and Sur, 2019*; *Liu et al., 2017*; *Reimer et al., 2016*). However, some (of these) studies also found unique contributions to pupil size in other subcortical regions, such as the cholinergic basal forebrain and dopaminergic midbrain, and the superior and inferior colliculi (*de Gee et al., 2017*; *Joshi et al., 2016*; *Mridha et al., 2019*; *Reimer et al., 2016*). Thus, the exact neuroanatomical and neurochemical source(s) of our observed effects of phasic arousal on decision-making remain to be determined.

There is mounting evidence for an arousal-linked reduction of biases and/or prior beliefs in humans, and our current findings generalize this emerging principle to rodents (mice) as well as to a higher-level form of decision-making that is based on information retrieved from memory. Previous work has shown a suppression of evidence accumulation bias similar to that which we found here during challenging visual perceptual choice tasks (contrast detection and random dot motion discrimination) (*de Gee et al., 2017*). Similarly, during sound localization in a dynamic environment, phasic arousal tracks a reduced influence of prior expectations on perception (*Krishnamurthy et al., 2017*). Furthermore, suppressive effects of phasic arousal also apply to choice history biases that evolve across trials. In this case, phasic arousal reflects perceptual decision uncertainty in the current trial and a reduction of choice repetition bias in the next trial (*Urai et al., 2017*). One study reported that phasic arousal tracked an overall reduction of reaction time during random dot motion discrimination (*van Kempen et al., 2019*). However, in this study, the signal strength was so high that the task could be solved without the temporal accumulation of evidence: reaction times were overall short (median <600 ms) and performance was close to ceiling. Thus, the arousal-dependent bias suppression seems to be a principle that generalizes across species, directions of bias, and domains of decision-making.

Some studies found a relationship between pupil responses and attention (*Binda et al., 2013*; *Ebitz and Moore, 2017*; *Naber et al., 2013*). In our tasks, this would predict a relationship between pupil-linked arousal and participants' sensitivity, RT or drift rate. But pupil responses were not systematically associated either of these measures (*Figures 1–3* and *Figure 4—figure supplement 2C*). A possible explanation for this discrepancy is that the studies discussed above focused on visual stimulus-evoked pupil responses, whereas we focused on the non-luminance-mediated, arousal-related pupil responses (*Joshi and Gold, 2020*; *Larsen and Waters, 2018*; *McGinley et al., 2015b*).

The use of the drift diffusion model in the current study was based on four main assumptions. First, in the go/no-go task, we assume that participants accumulated the auditory evidence *within*

each trial (discrete noise token) during a mini block and reset this accumulation process before the next discrete sound. Second, in both the go/no-go and yes/no tasks, we assume that subjects actively accumulated evidence towards both yes and no choices, which is supported by neurophysiological data from yes/no tasks (*Deco et al., 2007*; *Donner et al., 2009*). Third, in the go/no-go task, we assume that subjects set an implicit boundary for no choices (*Ratcliff et al., 2018*). Fourth, in the picture recognition task, we assume that memory-based decisions follow the same sequential sample principle established for perceptual decisions, whereby the 'samples' that are accumulated into the decision variable are drawn from memory (*Bowen et al., 2016*; *Ratcliff, 1978*; *Shadlen and Shohamy, 2016*). Although the quality of our model fits suggest that the model successfully accounted for the measured behavior, lending support to the validity of these assumptions, further work is needed to establish the evidence accumulation principle across decision tasks. This holds in particular for the memory-based decisions, where the hypothetical evidence 'samples' can neither be directly controlled nor observed.

The monotonic effects of 'phasic' arousal on the decision biases that we report here contrast with the observed effects of 'tonic' (pre-stimulus) arousal, which has a non-monotonic (inverted U) effect on behavior (perceptual sensitivity and bias) and neural activity (the signal-to-noise ratio of thalamic and cortical sensory responses) (*Gelbard-Sagiv et al., 2018*; *McGinley et al., 2015a*). Our study allows a direct comparison of tonic and phasic arousal effects within the same data set in mice. A previous report on that data set showed that the mice's behavioral performance was most rapid and accurate, and the least biased, at intermediate levels of 'tonic' arousal (medium baseline pupil size) (*McGinley et al., 2015a*). By contrast, we show here that the behavioral performance of the mice was linearly related to phasic arousal, with the most rapid and accurate and the least biased choices being associated with the most rapid phasic arousal responses. It is tempting to speculate that these differences result from different neuromodulatory systems governing tonic and phasic arousal. Indeed, rapid dilations of the pupil (phasic arousal) are more tightly associated with phasic activity in noradrenergic axons, whereas slow changes in pupil (tonic arousal) are accompanied by sustained activity in cholinergic axons (*Reimer et al., 2016*). Future physiological work should dissect the role of phasic and tonic neuromodulatory signals in decision-making computations in the brain.

Recent findings indicate that intrinsic behavioral variability is increased during sustained ('tonic') elevation of noradrenaline (NA) levels, in line with the 'adaptive gain theory' (*Aston-Jones and Cohen, 2005*). First, optical stimulation of LC inputs to anterior cingulate cortex caused rats to abandon strategic counter prediction in favor of stochastic choice in a competitive game (*Tervo et al., 2014*). Second, chemogenetic stimulation of the LC in rats performing a patch-leaving task increased decision noise and subsequent exploration (*Kane et al., 2017*). Third, pharmacologically reducing central NA levels in monkeys performing an operant effort exertion task parametrically increased choice consistency (*Jahn et al., 2018*). Finally, pharmacologically increasing central tonic NA levels in human subjects boosted the rate of alternations in a bistable visual input and long-range correlations in brain activity (*Pfeffer et al., 2018*). In the drift diffusion model, increased within-trial decision noise manifested as a decrease of the mean drift rate (quantifying the signal-to-noise ratio of decision evidence), which was not consistently evident in the present data. This is another indication, together with the baseline pupil effects reported by *McGinley et al., 2015a*, that the effects of phasic and tonic neuromodulation are distinct.

One influential account holds that phasic LC responses during decision-making are triggered by the threshold crossing in some circuit accumulating evidence, and that the resulting NA release then facilitates the translation of the choice into a motor act (*Aston-Jones and Cohen, 2005*). Within the drift diffusion model, this predicts a reduction in non-decision time and no effect on evidence accumulation. In contrast to this prediction, we found in all our datasets that phasic arousal affected evidence accumulation (suppressing biases therein), but not non-decision time. Our approach does not enable us to rule out an effect of phasic arousal on movement execution (i.e., kinematics). Yet, our results clearly establish an important role of phasic arousal in evidence accumulation, ruling out any *purely* post-decisional account. This implies that phasic LC responses driving pupil dilation are already recruited during evidence accumulation, or that the effect of pupil-linked arousal on evidence accumulation are governed by systems other than the LC. Future experiments characterizing phasic activity in the LC or in other brainstem nuclei that are involved in arousal during protracted evidence accumulation tasks could shed light on this issue.

Anatomical evidence supports the speculation that task-evoked neuromodulatory responses and cortical decision circuits interact in a recurrent fashion. One possibility is that neuromodulatory responses alter the balance between 'bottom-up' and 'top-down' signaling across the cortical hierarchy (*Friston, 2010*; *Hasselmo, 2006*; *Hsieh et al., 2000*; *Kimura et al., 1999*; *Kobayashi et al., 2000*). Sensory cortical regions encode likelihood signals and send these (bottom-up) to association cortex; participants' prior beliefs (about target probability, for example) are sent back (top-down) to the lower levels of the hierarchy (*Beck et al., 2012*; *Pouget et al., 2013*). Neuromodulators might reduce the weight of this prior data in the inference process (*Friston, 2010*; *Moran et al., 2013*), thereby reducing choice biases. Another possibility is neuromodulator release might scale with uncertainty about the incoming sensory data (*Friston, 2010*; *Moran et al., 2013*). Such a process could be implemented as top-down control by the cortical systems that are involved in decision-making over neuromodulatory brainstem centers. This line of reasoning is consistent with anatomical connectivity (*Aston-Jones and Cohen, 2005*; *Sara, 2009*). Finally, a related conceptual model that has been proposed for phasic LC responses is that cortical regions driving the LC (e.g. ACC) continuously compute the ratio of the posterior probability of the state of the world to its (estimated) prior probability (*Dayan and Yu, 2006*). LC is then activated when neural activity ramps towards the non-default choice (against ones' bias). The resulting LC activity might reset its cortical target circuits (*Bouret and Sara, 2005*) and override the default state (*Dayan and Yu, 2006*), facilitating the transition of the cortical decision circuitry towards the non-default state. These scenarios are in line with recent insights that (LC-mediated) pupil-linked phasic arousal shapes brain-wide connectivity (*Shine, 2019*; *Stitt et al., 2018*; *Zerbi et al., 2019*).

Our study showcases the value of comparative experiments in humans and non-human species. One would expect the basic functions of arousal systems (e.g. the LC-NA system) to be analogous in humans and rodents. Yet, it has been unclear whether these systems are recruited in the same way during decision-making. Computational variables such as decision uncertainty or surprise are encoded in prefrontal cortical regions (e.g. anterior cingulate or orbitofrontal cortex; *Kepecs et al., 2008*; *Ma and Jazayeri, 2014*; *Pouget et al., 2016*) and conveyed to brainstem arousal systems via top-down projections (*Aston-Jones and Cohen, 2005*; *Breton-Provencher and Sur, 2019*). Both the cortical representations of computational variables and the top-down projections to brainstem may differ between species. More importantly, it has not been known whether key components of the decision formation process, in particular evidence accumulation, would be affected by arousal signals in the same way in different species. Only recently has it been established that rodents (rats) and humans accumulate perceptual evidence in an analogous fashion (*Brunton et al., 2013*). Our results indicate that the shaping of evidence accumulation by phasic arousal is also governed by a principle that is conserved across species.

## Materials and methods

### Subjects

All procedures concerning the animal experiments were carried out in accordance with Yale University Institutional Animal Care and Use Committee, and are described in detail elsewhere (*McGinley et al., 2015a*). Human subjects were recruited and participated in the experiment in accordance with the ethics committee of the Department of Psychology at the University of Amsterdam (go/no-go and yes/no task), the ethics committee of Baylor College of Medicine (yes/no task with biased signal probabilities) or the ethics committee of the University of Hamburg (picture recognition task). Human subjects gave written informed consent and received credit points (go/no-go and yes/no tasks) or a performance-dependent monetary remuneration (yes/no task with biased signal probabilities and picture recognition task) for their participation. We analyzed two previously unpublished human data sets, and re-analyzed a previously published mice data set (*McGinley et al., 2015a*) and two human data sets (*Bergt et al., 2018*; *de Gee et al., 2017*). *Bergt et al., 2018* have analyzed pupil responses only during the encoding phase of the picture recognition memory experiment; we here present the first analyses of pupil responses during the recognition phase.

The sample sizes (and trial numbers per individual) for the newly collected human data sets were determined on the basis of the effects observed in a previous study comparing diffusion model

parameters between pupil conditions, with N = 14 (*de Gee et al., 2017*). The previously published mouse data set (*McGinley et al., 2015a*) consisted of data from five mice. Please note that the small number of mice was compensated by a substantial number of trials per individual, which is ideal for the detailed behavioral modeling we pursued here. We selected the data set by *Bergt et al., 2018* for across-subject correlations of variables of interest (*Figure 3*). This data set had a sufficient sample size for such an analysis, as determined on the basis of the effect size obtained in a previous study (*de Gee et al., 2014*).

Five mice (all males; age range, 2–4 months) and 20 human subjects (15 females; age range, 19–28 y) performed the go/no-go task. Twenty-four human subjects (of which 18 had already participated in the go/no-go task; 20 females; age range, 19–28 y) performed an additional yes/no task. Fifteen human subjects (eight females; age range, 20–28 y) performed the yes/no task with biased signal probabilities. Fifty-four human subjects (27 females; age range, 18–35 y) performed a picture recognition task, of which two were excluded from the analyses because of eye-tracking failure.

For the go/no-go task, mice performed between five and seven sessions (described in *McGinley et al., 2015a*), yielding a total of 2469–3479 trials per subject. For the go/no-go task, human participants performed 11 blocks of 60 trials each (distributed over two measurement sessions), yielding a total of 660 trials per participant. For the yes/no task, human participants performed between 11 and 13 blocks of 120 trials each (distributed over two measurement sessions), yielding a total of 1320–1560 trials per participant. For the yes/no task with biased signal probabilities, human subjects performed eight blocks of 120 trials each (distributed over two measurement sessions), yielding a total of 960 trials per participant. For the picture recognition task, human subjects performed 300 trials.

## Behavioral tasks

### Perceptual go/no-go auditory tone-in-noise detection task

Each mini block consisted of two to seven consecutive trials. Each trial was a distinct auditory noise stimulus of 1 s duration, and the inter-trial interval was 0.5 s. A weak signal tone was added to the last trial in the mini block (*Figure 1A*). The number of trials, and thus the signal position in the sequence, was randomly drawn beforehand. The probability of a target signal decreased linearly with trial number (*Figure 1—figure supplement 1A*, left). Each mini block was terminated by the subject's go response (hit or false alarm) or after a no-go error (miss). Each trial consisted of an auditory noise stimulus, or a pure tone added to one of the noise stimuli (cosine-gated 2 kHz for humans; new tone frequency each session for mice to avoid cortical reorganization across the weeks of training [*McGinley et al., 2015a*]). Noise stimuli were temporally orthogonal ripple combinations, which have spectro-temporal content that is highly dynamic, thus requiring temporal integration of the acoustic information in order to detect the stable signal tones (*McGinley et al., 2015a*). In the mouse experiments, auditory stimuli were presented at an overall intensity of 55 dB (root-mean-square [RMS] for each 1 s trial). In the human experiments, auditory stimuli were presented at an intensity of 65 dB using an IMG Stageline MD-5000DR over-ear headphone.

Mice learned to respond during the signal-plus-noise trials and to withhold responses during noise trials across training sessions. Mice responded by licking for sugar water reward. Human participants were instructed to press a button with their right index finger. Correct yes choices (hits) were followed by positive feedback: 4 μL of sugar water in the mice experiment, and a green fixation dot in the human experiment. In both mice and humans, false alarms were followed by an 8 s timeout. Humans, but not mice, also received an 8 s timeout after misses. This design difference was introduced to compensate for differences in general response bias between species that was evident in pilot experiments: mice tended to lick too frequently without a selective penalty for false alarms (i.e. liberal bias), whereas human participants exhibited a generally conservative intrinsic bias with balanced penalties for false alarms and correct rejects. Selectively penalizing false alarms would have aggravated this conservative tendency in humans, hence undermining the cross-species comparison of behavior.

The signal loudness was varied from trial to trial (−30 dB to 0 dB with respect to RMS noise in mice; −40 dB to –5 dB with respect to RMS noise in humans), while the 1-second mean RMS loudness of the noise was held constant. For the trial containing a signal tone within each mini block, signal loudness was selected randomly under the constraint that each of six (mice) or five (humans)

levels would occur equally often within each session (mice) or block of 60 mini blocks (humans). The corresponding signal loudness exhibited a robust effect on mean accuracy, with the highest accuracy for the loudest signal level: F(5,20) = 23.95 (p<0.001) and F(4,76) = 340.9 (p<0.001), for mouse and human subjects, respectively. Human hit rates were almost at ceiling level for the loudest signal (94.7% ± 0.69% s.e.m.), and close to ceiling for the second loudest signal (92.8% ± 0.35% s.e.m.). Because so few errors are not enough to constrain the drift diffusion model sufficiently, we merged the two conditions with the loudest signals.

## Perceptual yes/no (forced choice) auditory tone-in-noise detection task

Each trial consisted of two consecutive intervals (*Figure 2A*): (i) the baseline interval (3–4 s uniformly distributed); and (ii) the decision interval, the start of which was signaled by the onset of the auditory stimulus and which was terminated by the subject's response (or after a maximum duration of 2.5 s). The decision interval consisted of only an auditory noise stimulus (*McGinley et al., 2015a*), or a pure tone (2 kHz) superimposed onto the noise. In the first experiment, the signal was presented in 50% of trials. Auditory stimuli were presented at the same intensity of 65 dB using the same over-ear headphone as in the go/no-go task. In the second experiment, in order to manipulate perceptual choice bias experimentally, the signal was presented in either 30% of trials ('rare' blocks) or 70% of trials ('frequent' blocks) (*Figure 2B*). Auditory stimuli were presented at approximately the same signal loudness (65 dB) using a Sennheiser HD 660 s over-ear headphone, suppressing ambient noise.

Participants were instructed to report the presence or absence of the signal by pressing one of two response buttons with their left or right index finger, once they felt sufficiently certain (free response paradigm). The mapping between perceptual choice and button press (e.g., 'yes' → right key; 'no' → left key) was counterbalanced across participants. After every 40 trials, subjects were informed about their performance. In the second experiment, subjects were explicitly informed about signal probability. The order of signal probability (e.g., first 480 trials → 30%; last 480 trials → 70%) was counterbalanced across subjects.

Throughout the experiment, the target signal loudness was fixed at a level that yielded about 75% correct choices in the 50% signal probability condition. Each participant's individual signal, loudness was determined before the main experiment using an adaptive staircase procedure (Quest). For this, we used a two-interval forced choice variant of the tone-in-noise detection yes/no task (one interval, signal-plus-noise; the other, noise), in order to minimize contamination of the staircase by individual bias (generally smaller in two-interval forced choice than in yes/no tasks). In the first experiment, the resulting threshold signal loudness produced a mean accuracy of 74.14% correct (±0.75% s.e.m.). In the second experiment, the resulting threshold signal loudness produced a mean accuracy of 84.40% correct (±1.75% s.e.m.) and 83.37% correct (±1.36% s.e.m.) in the rare and frequent conditions, respectively. This increased accuracy was expected given the subjects' ability to incorporate prior knowledge about signal probability into their decision-making.

## Memory-based (forced choice) yes/no picture recognition decision task

The full experiment consisted of a picture and word encoding task, and a 24 hr-delayed free recall and recognition test (*Figure 3A*) previously described in *Bergt et al., 2018* (data available at https://figshare.com/articles/Reading_memory_formation_from_the_eyes/11432118). Here, we did not analyze data from the word recognition task because of a modality mismatch: auditory during encoding, visual during recognition. During encoding, 75 neutral and 75 negative grayscale pictures (modified to have the same average luminance) were randomly chosen from the picture pool (*Bergt et al., 2018*) and presented in randomized order for 3 s at the center of the screen, against a gray background that was equiluminant to the pictures. Subjects were instructed to memorize the pictures (intentional encoding) and to evaluate how emotional each picture was on a 4-point scale from 0 ('neutral') to 3 ('very negative'). During recognition, 24 hr post encoding, subjects saw all of the pictures that were presented on the first day and an equal number of novel neutral and negative items in randomized order. Subjects were instructed to indicate for each item whether it had been presented the previous day ('yes – old') or not ('no – new'). For items that were identified as 'old', participants were further asked to rate on a scale from 1 ('not certain') to 4 ('very certain') how confident they were that the item was indeed 'old'.

## Data acquisition

The mouse pupil data acquisition is described elsewhere (*McGinley et al., 2015a*). The human experiments were conducted in a psychophysics laboratory (go/no-go and yes/no tasks). The left eye's pupil was tracked at 1000 Hz with an average spatial resolution of 15 to 30 min arc, using an EyeLink 1000 Long Range Mount (SR Research, Osgoode, Ontario, Canada), and it was calibrated once at the start of each block.

## Analysis of task-evoked pupil responses

### Preprocessing

Periods of blinks and saccades were detected using the manufacturer's standard algorithms with default settings. The remaining data analyses were performed using custom-made Python scripts. We applied to each pupil timeseries: (i) linear interpolation of missing data due to blinks or other reasons (interpolation time window, from 150 ms before until 150 ms after missing data), (ii) low-pass filtering (third-order Butterworth, cut-off: 6 Hz), (iii) for human pupil data, removal of pupil responses to blinks and to saccades, by first estimating these responses by means of deconvolution and then removing them from the pupil time series by means of multiple linear regression (*Knapen et al., 2016*), (iv) conversion to units of modulation (percent signal change) around the mean of the pupil time series from each measurement session, and (v) down-sampling to 50 Hz. We computed the first time derivative of the pupil size, by subtracting the size from adjacent frames, and low-pass filtered the result (third-order Butterworth, cut-off: 2 Hz).

### Quantification of task-evoked pupil responses

In our previous work, which we aimed to extend here, we computed pupil responses aligned to subjects' behavioral choice (rather than stimulus onset), which was motivated by both theoretical considerations and empirical observations (*de Gee et al., 2014*; *de Gee et al., 2017*). All yes/no tasks (detection and recognition) were analogous in structure to the tasks from those previous studies. Thus, for those tasks, we here again computed task-evoked pupil responses time-locked to the behavioral choice (button press). Specifically, we computed pupil responses as the 95th percentile of the pupil derivative time series (*Reimer et al., 2016*) in the 500 ms before button press (gray windows in *Figure 2D* and *3B*). The resulting pupil bins were associated with different overall pupil response amplitudes (with regard to pre-trial baseline pupil size) across the whole duration of the trial (*Figure 2—figure supplement 1C* and *Figure 3—figure supplement 1B*).

The auditory go/no-go task entailed several deviations from the above task structure that required a different quantification of task-evoked pupil responses. The go/no task had, by design, an imbalance of motor responses between trials ending with different decisions, with no motor response for (implicit) no choices. Thus, no response-locking was possible for no-decisions, forcing us to deviate from our standard, choice-locked analysis approach, and to use stimulus-locking instead. Because decision times were substantially shorter in the go-/no-go tasks than in all the yes/no tasks (compare *Figure 1—figure supplement 1C* with *Figure 2—figure supplement 1B* and *Figure 3—figure supplement 1A*), behavioral correlates of pupil dilation should be less dependent on whether the pupil responses are aligned to stimulus onset or choice in the former.

In the go/no-go task, a transient drive of pupil dilation by the motor response (lick or button press) would yield larger (and potentially more rapid) pupil responses for go choices (motor movement) than for implicit no-go choices (no motor movement), even without any link between phasic arousal and decision bias. We took two approaches to minimize contamination by this motor imbalance. First, we quantified the single-trial response rate as the 95th percentile of the pupil derivative in an early window, ranging from the start of the period when the trial-average pupil derivative time course was significantly different from zero up to the first peak (gray windows in *Figure 1C*). For the mice, this window ranged from 40 to 230 ms after trial onset; for humans, this window ranged from 230 to 500 ms after trial onset. Second, we excluded decision intervals with a motor response before the end of this window plus a 50 ms buffer (cutoff 280 ms for mice, 550 ms for humans; *Figure 1—figure supplement 1C*). In both species, the resulting pupil-derivate-defined bins were associated with different overall pupil response amplitudes across the whole duration of the trial (*Figure 1—figure supplement 1G*).

Re-analyzing all yes/no data by time-locking the pupil responses to the stimulus onsets yielded pupil response time courses that were similar overall to the choice-aligned ones shown in the paper (Figure R1A). Yet, stimulus-locking did not yield a consistent relationship to any behavioral measure (RT, sensitivity/d′, or bias/criterion; Figure R1B-D), corroborating our use of choice-aligned pupil responses for inferring the role of phasic arousal signals in decision-making and evidence accumulation.

For analyses of the go/no-go task and the yes/no task with equal probability of signal occurrence, we used five equally populated bins of task-evoked pupil responses. As sensitivity increased with tone loudness (*Figure 1—figure supplement 1D*), there were more go-trials for louder tones (mean number of go-trials across subjects from least to most loud tones): 68, 73, 97, 122, 136 and 166 go-trials for mice and 45, 69, 89 and215 go-trials for human subjects. To ensure an equal number of tones of each loudness in each pupil bin, we binned trials by pupil response *within* a given loudness condition. We used three bins for the yes/no task with biased environments, because subjects performed substantially fewer trials (see '*Subjects*'). We used two bins for the picture recognition task, so that we could perform the individual difference analysis reported in *Figure 3*. In the picture recognition task, we ensured that each pupil bin contained an equal number of neutral and emotional stimuli. In all cases, the results are qualitatively the same when using five equally populated bins of task-evoked pupil responses.

## Analysis and modeling of choice behavior

In the go/no-go task, the first trial of each mini block (see '*Behavioral tasks*') was excluded from the analyses, because this trial served as a mini block start cue, and never included the signal (pure sine wave). In the go/no-go and yes/no tasks, reaction time (RT) was defined as the time from stimulus onset until the lick or button press. In the mice go/no-go data set, trials with RTs shorter than 280 ms were excluded from the analyses (see '*Quantification of task-evoked pupillary responses*' and *Figure 1—figure supplement 1C*); in all human data sets, trials with RTs shorter than 550 ms were excluded from the analyses (*Figure 1—figure supplement 1C*, *Figure 2—figure supplement 1B* and *Figure 3—figure supplement 1A*).

### Signal-detection theoretic modeling (go/no-go and yes/no tasks)

The signal detection theoretic (SDT) metrics sensitivity (d′) and criterion (c) (*Green and Swets, 1966*) were computed separately for each of the bins of pupil response size. We estimated sensitivity as the difference between z-scores of hit rates and false-alarm rates. We estimated criterion by averaging the z-scores of hit rates and false-alarm rates and multiplying the result by −1.

In the go/no-go task, subjects could set only one decision criterion (not to be confused with the above-defined c), against which to compare sensory evidence so as to determine choice. This is because signal loudness was drawn pseudo-randomly for each trial and participants had no way of using separate criteria for different signal strengths. We reconstructed this overall decision criterion (irrespective of signal loudness) and used this as a measure of the overall choice bias, whose dependence on pupil response we then assessed (*Figure 1D*). To this end, we used the following approach derived from SDT (*Green and Swets, 1966*). We computed one false alarm rate (based on the noise trials) and multiple hit rates (one per signal loudness). Based on these, we modeled one overall noise distribution (normally distributed with mean = 0, sigma = 1), and one 'composite' signal distribution (*Figure 1—figure supplement 1B*), which was computed as the average across a number of signal distributions separately modeled for each signal loudness (each normally distributed with mean = empirical d′ for that signal loudness, and sigma = 1).

We defined the 'zero-bias point' (*Z*) as the value for which the noise and composite signal distributions crossed:

$$S(Z) - N(Z) = 0 \tag{1}$$

where *S* and *N* are the composite signal and noise distributions, respectively.

The subject's empirical 'choice point' (*C*) was computed as:

$$C = \left(0.5 \times d_i'\right) + c_i \tag{2}$$

where $d_i$ and $c_i$ were a subject's SDT-sensitivity and SDT-criterion for a given signal loudness, 'i'. Note that C is the same constant when sensitivity and criterion are computed for each signal loudness based on the same false alarm rate. Therefore, it does not matter which signal loudness is used to compute the empirical choice point.

Finally, the overall bias measure was then taken as the distance between the subject's choice point and the zero-bias point:

$$\text{Overall bias} = C - Z \tag{3}$$

## Drift diffusion modeling

Data from all tasks were fit with the drift diffusion model, which well captured all of the features of behavior that we assessed. We used the HDDM 0.7.5 package (*Wiecki et al., 2013*) to fit behavioral data from the yes/no and go/no-go tasks. In all datasets, we allowed the following parameters to vary with pupil response-bins: (i) the separation between both bounds (i.e. response caution); (ii) the mean drift rate across trials; (iii) drift bias (an evidence independent constant added to the drift); and (iv) the non-decision time (sum of the latencies for sensory encoding and motor execution of the choice). In the datasets using yes/no protocols, we additionally allowed starting point to vary with pupil response bin. In the go/no-go datasets, we allowed non-decision time, drift rate, and drift bias to vary with signal strength (i.e., signal loudness). The specifics of the fitting procedures for the yes/no and go/no-go protocols are described below.

To verify that the best-fitting models did indeed account for the pupil response-dependent changes in behavior (blue X's in *Figures 1–3*), we generated a simulated data set using the fitted drift diffusion model parameters. Separately for each subject, we simulated 100K trials, while ensuring that the fraction of signal+noise vs noise trials matched that of the empirical data; we then computed RT, as well as sensitivity (signal detection d') and overall bias (for the go/no-go data sets) or criterion (for the rest) for every bin (as described above).

We used a similar approach to test which of the models in *Figure 6A* best explained the pupil response-dependent changes in behavior (X's in *Figure 6B*). Separately for each subject, we simulated data (100K trials per subject) using the fitted parameters of the three alternative models. We then computed choice bias (*Figure 6B*), RT (*Figure 6—figure supplement 1B*), and sensitivity (*Figure 6—figure supplement 1C*). Separately for each model, we computed the residuals (sum of squares of differences) between the empirical and model-predicted choice metrics (*Figure 6C*).

We also used simulations to test whether systematic variations in drift bias would appear as random trial-to-trial variability in the accumulation process (drift rate variability) (*Figure 5*). We simulated one million trials from two conditions that differed according drift bias (−0.5 in condition 1; 0 in condition 2). Drift rate, boundary separation and non-decision time were fixed at 1, 1, and 0.3, respectively; drift rate variability was fixed to 0. We then fitted the model to the simulated data (with the G square method; see below), letting only drift bias vary with condition, and to another version of the model in which we fixed drift bias across the two conditions. We repeated the same procedure for a number of drift bias differences between the simulated conditions (*Figure 5B*).

## Yes-no task

We fitted all yes/no datasets using Markov-chain Monte Carlo sampling as implemented in the HDDM toolbox (*Wiecki et al., 2013*). Fitting the model to RT distributions for the separate responses (termed 'stimulus coding' in *Wiecki et al., 2013*) enabled the estimation of parameters that could have induced biases towards specific choices. Bayesian MCMC generates full posterior distributions over parameter estimates, quantifying not only the most likely parameter value but also the uncertainty associated with that estimate. The hierarchical nature of the model assumes that all observers in a data set are drawn from a group, with specific group-level prior distributions that are informed by the literature. In practice, this results in more stable parameter estimates for individual subjects, who are constrained by the group-level inference. The hierarchical nature of the model also minimizes the risk of overfitting the data (*Katahira, 2016*; *Vandekerckhove et al., 2011*; *Wiecki et al., 2013*). Together, this allowed us to vary all of the main parameters simultaneously with pupil bin (so that they could 'compete' capturing the pupil-linked reduction of choice bias): starting point, boundary separation, drift rate, drift bias and non-decision time. We ran three separate Markov chains with 12,500 samples each. Of those, 2500 were discarded as burn-in.

Individual parameter estimates were then estimated from the posterior distributions across the resulting 10,000 samples. All group-level chains were visually inspected to ensure convergence. In addition, we computed the Gelman-Rubin $\hat{R}$ statistic (which compares within-chain and between-chain variance) and checked that all group-level parameters had an $\hat{R}$ between 0.99 and 1.01.

To test the robustness of the significance of the pupil-dependent effect on drift bias, we re-fitted the full model, but now fixing drift criterion or starting point with pupil response bin (*Figure 4—figure supplement 1C*). Using deviance information criterion (*Spiegelhalter et al., 2002*) for model selection, we compared whether the added complexity of our original model was justified to account for the data. This is a common metric for comparing hierarchical models, for which a unique 'likelihood' is not defined and the effective number of degrees of freedom is often unclear (*Spiegelhalter et al., 2002*).

The models in *Figure 6* were fitted on the basis of continuous maximum likelihood using the Python-package PyDDM (*Shinn et al., 2020*). We modeled urgency as hyperbolically collapsing bounds (*Urai et al., 2019*). Specifically, the two choice bounds were modeled to vary over time as follows:

$$
\begin{aligned}
\mathrm{a}_{up}(\mathrm{t}) &= \left| a - a\frac{t}{t+c} \right|_{a/2}^{a} \\
\mathrm{a}_{down}(\mathrm{t}) &= \left| a\frac{t}{t+c} \right|_{0}^{a/2}
\end{aligned}
\tag{4}
$$

In the above, the notation $|x|_{\min}^{\max}$ indicates that $x$ was clamped such that $x \in [\min, \max]$.

## Go/no-go task

The above-described hierarchical Bayesian fitting procedure was not used for the go/no-go tasks because a modified likelihood function was not yet successfully implemented in HDDM. Instead, we fitted the go/no-go data on the basis of RT quantiles, using the so-called G square method (code contributed to the master HDDM repository on Github: https://github.com/hddm-devs/hddm/blob/master/hddm/examples/gonogo_demo.ipynb). The RT distributions for yes choices were represented by the 0.1, 0.3, 0.5, 0.7 and 0.9 quantiles, and, along with the associated response proportions, contributed to G square; a single bin containing the number of no-go choices contributed to G square (*Ratcliff et al., 2018*). Starting point and drift rate variability were fitted but fixed across the pupil-defined bins. In addition, drift rate, drift bias and non-decision time varied with signal loudness. The same noise-only trials were re-used when fitting the model to each signal loudness.

The absence of no-responses in the go/no-go protocol required fixing one of the two bias parameters (starting point or drift bias) as a function of pupil response; leaving both parameters free to vary led to poor parameter recovery. We fixed starting point on the basis of formal model comparison between a model with pupil-dependent variation of drift bias and starting point (*Figure 4—figure supplement 1B*). For simplicity, here we ignored signal loudness, but the same was true when explicitly modeling signal loudness (data not shown).

## Statistical comparisons

We used a mixed linear modeling approach implemented in the Python-package '*Statsmodels*' (*Seabold and Perktold, 2010*) to quantify the dependence of several metrics of overt behavior, or of estimated model parameters (see above), on pupil response bin. For the go/no-go task, we simultaneously quantified the dependence on signal loudness. Our approach was analogous to sequential polynomial regression analysis (*Draper and Smith, 1998*), but now performed within a mixed linear modeling framework. In the first step, we fitted two mixed models to test whether pupil response bin predominantly exhibited a monotonic effect (first-order) or a non-monotonic effect (second-order) on the behavioral metric of interest (*y*). The fixed effects were specified as:

$$
\begin{aligned}
\text{Model 1: } \mathrm{y} &\sim \beta_0 1 + \beta_1 S + \beta_2 P^1 \\
\text{Model 2: } \mathrm{y} &\sim \beta_0 1 + \beta_1 S + \beta_2 P^1 + \beta_3 P^2
\end{aligned}
\tag{5}
$$

with $\beta$ as regression coefficients, $S$ as the signal loudness (for go/no-go task), and $P$ as the task-evoked pupil response bin number. We included the maximal random effects structure justified by the design (*Barr et al., 2013*). For data from the go/no-go task, that meant that intercepts and signal loudness and pupil bin coefficients could vary with participant. For data from the yes/no tasks,

intercepts and pupil bin coefficients could vary with participant. The mixed models were fitted through restricted maximum likelihood estimation. Each model was then sequentially tested in a serial hierarchical analysis, which was based on BIC. This analysis was performed for the complete sample at once, and it tested whether adding the next higher order model yielded a significantly better description of the response than the respective lower order model. We tested models from the first-order (constant, no effect of pupil response) up to the second-order (quadratic, non-monotonic). In the second step, we refitted the winning model through restricted maximum likelihood estimation (REML).

We used paired-sample t-tests to test for significant differences between the pupil derivative time course and 0, between pupil responses for yes versus no choices, and for behavioral metrics between pupil response bins in the picture recognition experiment.

### Data and code sharing

The data and analysis scripts are publicly available on https://github.com/jwdegee/2020_eLife_pupil_bias (copy archived at https://github.com/elifesciences-publications/2020_eLife_pupil_bias; *de Gee, 2020*).

## Acknowledgements

We thank Daniëlle Rijkmans, Guusje Boomgaard and Christopher David Riddell for help with the data collection for the human auditory detection tasks, Michael Beauchamp and colleagues at the Core for Advanced Magnetic Resonance Imaging at Baylor College of Medicine for letting us use the psychophysics lab to collect data for the human auditory detection tasks with biased environments, Anne Bergt for help with the data collection for the human picture recognition task, and all members of the Donner lab for discussion. This research was supported by the German Research Foundation (DFG, grant numbers: DO 1240/3–1 and SFB 936A7 to THD), the European Commission CH2020 7[th] Framework Programme (Marie Skłodowska-Curie Individual Fellowship: 658581-CODIR, to KT and THD), and the National Institutes of Health (R03DC015618 and R01DC017797, to MJM).

## Additional information

### Competing interests

Tobias H Donner: Reviewing editor, *eLife*. The other authors declare that no competing interests exist.

### Funding

| Funder | Grant reference number | Author |
| --- | --- | --- |
| Deutsche Forschungsgemeinschaft | DO 1240/3-1 | Tobias H Donner |
| Deutsche Forschungsgemeinschaft | DO 1240/4–1 | Tobias H Donner |
| Deutsche Forschungsgemeinschaft | SFB 936A7 | Tobias H Donner |
| European Commission | 658581-CODIR | Konstantinos Tsetsos Tobias H Donner |
| National Institutes of Health | R03DC015618 | Matthew J McGinley |
| National Institutes of Health | R01DC017797 | Matthew J McGinley |
| German Academic Exchange Service | A/13/70362 | Anne E Urai |
| German National Academy of Sciences Leopoldina | LPDS 2019-03 | Anne E Urai |

The funders had no role in study design, data collection and interpretation, or the decision to submit the work for publication.

## Author contributions
Jan Willem de Gee, Conceptualization, Data curation, Formal analysis, Investigation, Visualization, Writing - original draft, Writing - review and editing; Konstantinos Tsetsos, Conceptualization, Methodology, Writing - original draft, Writing - review and editing; Lars Schwabe, David McCormick, Conceptualization, Writing - review and editing; Anne E Urai, Data curation, Writing - review and editing; Matthew J McGinley, Conceptualization, Methodology, Data curation, Supervision, Funding acquisition, Investigation, Writing - original draft, Writing - review and editing; Tobias H Donner, Conceptualization, Supervision, Funding acquisition, Writing - original draft, Project administration, Writing - review and editing

## Author ORCIDs
Jan Willem de Gee (iD) https://orcid.org/0000-0002-5875-8282
Konstantinos Tsetsos (iD) http://orcid.org/0000-0003-2709-7634
Lars Schwabe (iD) http://orcid.org/0000-0003-4429-4373
Anne E Urai (iD) http://orcid.org/0000-0001-5270-6513
Tobias H Donner (iD) https://orcid.org/0000-0002-7559-6019

## Ethics
Human subjects: All procedures concerning the animal experiments were carried out in accordance with Yale University Institutional Animal Care and Use Committee, and are described in detail elsewhere (McGinley, David, et al., 2015). Human subjects were recruited and participated in the experiment in accordance with the ethics committee of the Department of Psychology at the University of Amsterdam (go/no-go and yes/no task), the ethics committee of Baylor College of Medicine (yes/no task with biased signal probabilities) or the ethics committee of the University of Hamburg (picture recognition task). Human subjects gave written informed consent and received credit points (go/no-go and yes/no tasks) or a performance-dependent monetary remuneration (yes/no task with biased signal probabilities and picture recognition task) for their participation.

Animal experimentation: All procedures concerning the animal experiments were carried out in accordance with Yale University Institutional Animal Care and Use Committee, and are described in detail elsewhere (McGinley, David, et al., 2015).

## Decision letter and Author response
Decision letter https://doi.org/10.7554/eLife.54014.sa1
Author response https://doi.org/10.7554/eLife.54014.sa2

# Additional files

## Supplementary files
• Transparent reporting form

## Data availability
The data and analysis scripts are publicly available on https://github.com/jwdegee/2020_eLife_pupil_bias (copy archived at https://github.com/elifesciences-publications/2020_eLife_pupil_bias).

The following previously published dataset was used:

| Author(s) | Year | Dataset title | Dataset URL | Database and Identifier |
|---|---|---|---|---|
| Bergt A, Urai AE, Donner TH, Schwabe L | 2018 | Reading memory formation from the eyes | https://doi.org/10.5281/zenodo.1246101 | Zenodo, 10.5281/zenodo.1246101 |

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
