## [Decision Letter]

Thank you for submitting your article "Phasic arousal suppresses biases in mice and humans across domains of decision-making" for consideration by *eLife*. Your article has been reviewed by three peer reviewers, and the evaluation has been overseen by a Reviewing Editor and Joshua Gold as the Senior Editor. The following individuals involved in review of your submission have agreed to reveal their identity: Marcus Grüschow (Reviewer #1); R Becket Ebitz (Reviewer #2).

The reviewers have discussed the reviews with one another and the Reviewing Editor has drafted this decision to help you prepare a revised submission.

Summary:

This study builds on a previous *eLife* paper ("Dynamic modulation of decision biases by brainstem arousal systems") that showing that phasic pupil responses predict reduced conservative accumulation biases in humans performing perceptual tasks. Here the authors show that the biases: 1) are reduced in both directions (liberal/conservative), 2) are comparable in both mice and humans, and 3) occur in both in memory-based and perceptual decisions There was general agreement that the new work is a useful and substantial extension of the previous work.

Essential revisions:

1) There is a consistent conflation of measured correlations/predictions with what seem to be much stronger claims about causality; e.g., in the title, along with many places in the text, that claim that "phasic arousal suppresses…" (or "reduces", "affected", etc). These points should be disentangled and clarified.

2) Several questions/concerns were noted regarding the DDM fits. In particular:

a) It seems that the core feature captured by the accumulation bias term in the DDM (the growing bias with longer RTs) might also be explained by a starting point bias in combination with an urgency signal (e.g., as bounds decay, the relative bias in the starting point becomes more and more important). Are the authors making a strong claim about the exact process model that they have chosen? And if so, could they rule out the alternative model above using model comparison?

b) Popular theories of arousal and decision-making argue that the role of enhanced arousal is to enhance the representation of task-relevant variables (Mather GANE), which would seem to argue for an enhanced drift rate under high arousal. It would thus be important to provide the model comparison (BIC, DIC) between the DDM provided by the authors and a model that does not contain the additive bias term, possibly with a fixed or variable starting point. To be more precise, traditionally, in the sequential sampling framework, response biases have been modeled as a shift in starting point. If one claims that response biases are implemented as a "drift criterion", it would be good to explicitly show that this is a better model of the data. Mulder et al., 2012 (JoN, their Figure 2) present a qualitative distinctive test for these two models: if the choice bias is implemented as a drift rate bias, the response time patterns for validly/invalidly biased trials should be the same for correct and incorrect responses. However, if the choice bias is implemented as a starting point bias, there should be opposite effect for correct and incorrect trials: for correct trials, invalidly biased trials should be slower than validly biased trials, whereas for incorrect trials, this pattern should reverse: invalidly biased trials should be *faster* than validly biased trials. Could the authors plot the averaged (over subjects) median response times of correct/incorrect x validly/invalidly biased trials for their second experiment, where subjects are biased due to the probabilistic nature of the environment, just as in Figure 2 of Mulder et al., 2012? Furthermore, for all experiments where there are two choice options (so not the go/no-go tasks, where the DDM is under-constrained), could they formally compare (a) a model where only starting point is allowed to implement response biases, (b) a model where only drift criterion is allowed to implement the response biases, and (c) a model where both starting point and drift criterion are allowed to implement the response biases?

3) Given that phasic pupil changes are sensitive to baseline pupil diameter, it would be useful to have more information about the statement "Variation in pre-trial pupil size causes floor and ceiling effects on phasic dilations, shaped by light conditions.” What exactly do the authors mean here? Where do the differential light conditions come from in auditory detection tasks? Moreover, how did you account for pre-trial spill-over effects on pupil; e.g., pre-trial-difficulty (loudness), response (yes/no), pupil dilation/derivative, tonic level, etc.

4) The choice of pupil change measure differs across tasks, due to differences in motor demands. However, every task has a stimulus onset and it does look like there were substantive differences in the stimulus-aligned pupil responses in all three experiments. It would be helpful to know if stimulus-aligned pupil responses are predictive in experiments 2 and 3. Our sense is that the attentional literature would likely predict changes in stimulus-aligned pupil responses, rather than motor aligned (and probably in stimulus-aligned dilation for auditory stimuli and stimulus-aligned constriction for visual stimuli).

Relatedly, many studies have looked at the relationship between stimulus attention and the transient pupil response to both visual and auditory stimuli, in both humans and non-human primates (in whom some mechanistic work has been done). This literature is not discussed here, which is striking given that it would seem that some of the results reported here explained by a decrease in attention paid to sensory evidence and previous studies have shown that attention can scale DDM drift.

5) We admire the combination of mice and men in this work however, the entire manuscript features only 5 mice in the first experiment. We believe it would be beneficial if the authors could justify this small sample size for other non-animal researchers.

6) It would be useful validate that the measure of bias in Figure 1 is not sensitive to the number of trials in each loudness condition, as this will clearly differ across pupil bins. We are not sure about the total number of trials in some of the loudness conditions, but we worry a bit that if there are very few trials in a given loudness condition, the measure could be misleading.

7) It would be useful to flesh out in a bit more detail why the observation of non-linear vs. linear arousal state modulation would indicate distinct functional roles for tonic vs. phasic arousal respectively. First, a linear relationship could just represent the increasing part of the inverted u-shape and secondly the original cited papers by Aston-Jones and Cohen, 2005; Yerkes and Dodson, 1908, used pupil size, while I believe the current work focuses on the speed of dilation. It would be important to clarify as to how these measures are related or how they may reflect activity of distinct neuromodulators or modulatory receptors as the authors pointed out in the final paragraph of subsection “In humans and mice, phasic arousal tracks a reduction of choice bias in an auditory detection task”.

8) The authors argue that experiment 3 shows reduced sampling from memory, but the idea that mnemonic decisions are based on samples from memory is still an early theory. Further, to our knowledge we don't know that the DDM does recover information about this memory sampling process (i.e. we do not know whether DDM drift biases reflect sampling from memory, sampling from the stimulus, or some other time-varying decisional process like response competition). It would be helpful to be more circumscribed in the interpretation of these results as reflecting a change in sampling from memory.

9) What distribution was used to determine the length of mini-blocks? The Materials and methods say that miniblocks were never more than 7 trials, which suggests that the hazard rate for a signal trial was not constant across the miniblock. Also, the hazard rate for the reference trial is presumably zero, were these trials included in analysis? Elsewhere the authors note that the hazard for signal trials was kept approximately flat. How approximately? Were phasic pupil responses related to changes in hazard across miniblocks?

---

## [Author Response]

Essential revisions:1) There is a consistent conflation of measured correlations/predictions with what seem to be much stronger claims about causality; e.g., in the title, along with many places in the text, that claim that "phasic arousal suppresses…" (or "reduces", "affected", etc). These points should be disentangled and clarified.

Thank you for pointing us to this important issue. We agree that our data does not allow for claims about causality, and we have now modified our language accordingly throughout all section of the paper, including the title.

2) Several questions/concerns were noted regarding the DDM fits. In particular:a) It seems that the core feature captured by the accumulation bias term in the DDM (the growing bias with longer RTs) might also be explained by a starting point bias in combination with an urgency signal (e.g., as bounds decay, the relative bias in the starting point becomes more and more important). Are the authors making a strong claim about the exact process model that they have chosen? And if so, could they rule out the alternative model above using model comparison?

Thank you for raising this issue. Indeed, we do not intend to make “strong claims” about the particular process model used, and we took care to present the scope of our modeling efforts as well as discussing the validity of the underlying assumptions (see dedicated paragraph in the Discussion, paragraph five) in an accurate way that does not oversell the results. That said, while certainly oversimplified, the DDM is useful for the specific purpose we use it for: differentiating between two distinct, but a priori equally possible, algorithmic schemes for implementing a choice bias.

We have now addressed the impact of a potential urgency signal on our inference on the underlying biasing mechanisms through fitting a new model combining starting point bias with collapsing bounds. This model failed to capture the observed pupil-linked changes in choice bias (see new Results subsection “Changes in urgency unlikely to mediate the pupil-linked reduction of choice bias” and new Figure 5). We hope that you will now find that our modeling results provide sufficient and compelling support for our main conclusion.

b) Popular theories of arousal and decision-making argue that the role of enhanced arousal is to enhance the representation of task-relevant variables (Mather GANE), which would seem to argue for an enhanced drift rate under high arousal.

In our original work on this issue (de Gee et al., 2014, 2017), we had indeed anticipated a relationship between phasic, pupil-linked arousal and sensitivity / mean drift rate. But this was not observed; instead we found a selective effect on bias / drift criterion. In our current work, we also assessed the relationship between pupil responses with markers of sensitivity to task-relevant information: sensitivity (d’), mean RT, and drift rate (Figures 1-4 and associated figure supplements) — we again did not observe any consistent relationship here. In the Results section, we have now added a paragraph to address this point explicitly.

It would thus be important to provide the model comparison (BIC, DIC) between the DDM provided by the authors and a model that does not contain the additive bias term, possibly with a fixed or variable starting point. To be more precise, traditionally, in the sequential sampling framework, response biases have been modeled as a shift in starting point. If one claims that response biases are implemented as a "drift criterion", it would be good to explicitly show that this is a better model of the data.

We now added the results of formal model comparisons between “drift criterion only” and “starting point only model”, shown in Figure 4—figure supplement 1. In short, the drift criterion model wins in all cases. This is described in paragraph four of subsection “Pupil-linked bias reduction is associated with changes in the evidence accumulation process”.

We are unsure about which traditional work your comment refers to. But we would like to point out that our modeling of bias within the DDM, in terms of either starting point or drift bias, is in line with a substantial body of influential papers in this field (e.g. (de Gee et al., 2017; Hanks et al., 2011; Moran, 2015; Ratcliff and McKoon, 2008; Urai et al., 2019).

Mulder et al., 2012 (JoN, their Figure 2) present a qualitative distinctive test for these two models: if the choice bias is implemented as a drift rate bias, the response time patterns for validly/invalidly biased trials should be the same for correct and incorrect responses. However, if the choice bias is implemented as a starting point bias, there should be opposite effect for correct and incorrect trials: for correct trials, invalidly biased trials should be slower than validly biased trials, whereas for incorrect trials, this pattern should reverse: invalidly biased trials should be *faster* than validly biased trials. Could the authors plot the averaged (over subjects) median response times of correct/incorrect x validly/invalidly biased trials for their second experiment, where subjects are biased due to the probabilistic nature of the environment, just as in Figure 2 of Mulder et al., 2012?

We fully agree with the benefit of showing qualitative assessments of the data to help differentiate between the starting point and drift bias models. We have now used so-called “conditional response functions" (White and Poldrack, 2014) to do that (shown in Figure 4—figure supplement 2). These qualitative tests are in line with our previously presented analyses, in indicating that the pupil-linked shift in choice bias is implemented by a shift in drift bias, not starting point.

Please note that our experiments did not allow for an exact replication of the analysis done by Mulder et al., 2012: This requires plotting error/correct RTs, separately for valid/neutral/invalid cues. First, they used a cuing paradigm with valid/neutral/invalid cues, which was not included in any of our tasks. Second, they define a starting point or drift bias as a shift towards the correct or incorrect choice boundary, while we have defined a starting point or drift bias as a shift towards the “yes” or “no” choice boundary, in line with the standard definition of bias within the DDM (e.g. (de Gee et al., 2017; Hanks et al., 2011; Moran, 2015; Ratcliff and McKoon, 2008; Urai et al., 2019).

Furthermore, for all experiments where there are two choice options (so not the go/no-go tasks, where the DDM is under-constrained), could they formally compare (a) a model where only starting point is allowed to implement response biases, (b) a model where only drift criterion is allowed to implement the response biases, and (c) a model where both starting point and drift criterion are allowed to implement the response biases?

We now added the results of formal model comparisons between “drift criterion only” and “starting point only models”, shown in Figure 4—figure supplement 1. In short, the drift criterion model wins in all cases. This is described in paragraph four of subsection “Pupil-linked bias reduction is associated with changes in the evidence accumulation process”.

As I final reply to this list of comments, we would like to express that critical experimental evidence is sorely needed in order to constrain the many current frameworks of the impact of arousal on cognition. In this case, we have found that changes in “drift bias” with phasic arousal indeed are a better model of the data than “starting point bias”. We see our role in helping to build such evidence, and this manuscript aims to make a humble contribution to this endeavor.

3) Given that phasic pupil changes are sensitive to baseline pupil diameter, it would be useful to have more information about the statement "Variation in pre-trial pupil size causes floor and ceiling effects on phasic dilations, shaped by light conditions.” What exactly do the authors mean here? Where do the differential light conditions come from in auditory detection tasks? Moreover, how did you account for pre-trial spill-over effects on pupil; e.g., pre-trial-difficulty (loudness), response (yes/no), pupil dilation/derivative, tonic level, etc.

We apologize for this confusion. The luminance was held constant in all tasks. We have now clarified our rationale in subsection “Suppression of bias by phasic arousal is distinct from ongoing slow state fluctuations”.

4) The choice of pupil change measure differs across tasks, due to differences in motor demands. However, every task has a stimulus onset and it does look like there were substantive differences in the stimulus-aligned pupil responses in all three experiments. It would be helpful to know if stimulus-aligned pupil responses are predictive in experiments 2 and 3. Our sense is that the attentional literature would likely predict changes in stimulus-aligned pupil responses, rather than motor aligned (and probably in stimulus-aligned dilation for auditory stimuli and stimulus-aligned constriction for visual stimuli).

We appreciate the point that aligning pupil responses to stimulus onset would allow for using the exact same pupil response measure across all data sets. However, our previous work on this topic (de Gee et al., 2014, 2017), which consistently found pupil-linked bias reductions and which we are extending here, was based on choice-aligned pupil responses. This was motivated by theoretical considerations and empirical findings. First, highly influential accounts of phasic neuromodulatory (locus coeruleus) activity in decision-making (Aston-Jones and Cohen, 2005; Dayan and Yu, 2006) have linked those responses to the decision process (in the case of Aston-Jones and Cohen even its termination through boundary crossing) rather than to stimulus onset; aligning pupil responses to choice allows for most directly relating our findings to these accounts. Second, our brainstem fMRI measurements (de Gee et al., 2017) showed clearer pupil-linked LC-responses when aligning both to the choice, rather than to stimulus onset. Since we are here using the pupil responses as a proxy of phasic brainstem activity, this observation also makes choice-locking more appropriate.

Now, choice-locking was obviously not possible in the go/no-go task (absence of no-choices) and the asymmetry in motor factors between the two choice conditions forced us to eliminate motor confounds as much as possible by the choice of analysis time window. These factors forced us to deviate from our standard, and generally preferred, choice-locking in the analysis of these data sets.

We have now re-analyzed all yes/no data by time-locking the pupil responses to the stimulus onsets (Author response image 1). This shows overall very similar pupil response time courses to the choice-aligned ones shown in the paper (Author response image 1), confirming your expectation that there are pupil dilations for auditory stimuli and pupil constriction for the visual stimuli in the recognition task (please note that the same is evident in the choice-locked pupil responses shown in Figures 1-3). Yet, we do not find any consistent relationships to any behavioral measure (RT, sensitivity / d’, or bias / criterion; Author response image 1), in line with the theoretical reasoning above and corroborating our use of choice-aligned pupil responses for inferring the role of phasic arousal signals in decision-making and evidence accumulation.

One may wonder why, then, we found any consistent effects in the stimulus-aligned analysis of the go/no-go data at all. One speculation is that, because decision times were substantially shorter than for all the yes/no tasks (compare Figure 1—figure supplement 1 with the corresponding supplements of Figures 2/3), any correlates of pupil dilation might be expected to be less dependent on whether the pupil responses are aligned to stimulus onset or choice.

**Author response image 1. sa2fig1:** Behavioral correlates of stimulus-locked pupil responses in yes/no tasks. (**A**) Task-evoked pupil response (solid line) and response derivative (dashed line) aligned to stimulus onset in the yes/no detection datasets with equal, rare, and frequent signals and the yes/no recognition dataset, respectively. Grey window, interval for task-evoked pupil response measures (range, 0.23-0.5 s); black bar, significant pupil derivative; stats, cluster-corrected one-sample t-test. (**B**) Relationship between RT and task-evoked pupil response. Linear fits were plotted if first-order fit was superior to constant fit; quadratic fits were not superior to first-order fits. Stats, mixed linear modeling (detection data sets), or paired-samples t-test (recognition data set). (**C**) As B, but for perceptual sensitivity. (**D**) As B, but for choice bias. All panels: group average (N = 24; N = 15; N = 15; N = 54); shading or error bars, s.e.m.

Relatedly, many studies have looked at the relationship between stimulus attention and the transient pupil response to both visual and auditory stimuli, in both humans and non-human primates (in whom some mechanistic work has been done). This literature is not discussed here, which is striking given that it would seem that some of the results reported here explained by a decrease in attention paid to sensory evidence and previous studies have shown that attention can scale DDM drift.

See our reply above regarding the lack of consistent association between pupil responses and parameters quantifying sensitivity (d’, RT, mean drift rate). (Please note that the same holds also for a different measure of the response amplitude, based on the commonly used baseline-corrected response, rather than the derivative used in the current study — see (de Gee et al., 2017)).

We now refer to papers on the relationship between attention and (typically luminance-mediated) pupil responses in a new Discussion paragraph.

5) We admire the combination of mice and men in this work however, the entire manuscript features only 5 mice in the first experiment. We believe it would be beneficial if the authors could justify this small sample size for other non-animal researchers.

We now write on: “The previously published mice data set (McGinley et al., 2015) consisted of five mice. Please note that the small number of mice was compensated by a substantial number of trials per individual, which is ideal for the detailed behavioral modeling we pursued here.”

This data set was the first published on pupil and rodent sensory signal detection. The relatively small sample size resulted from the fact that the pupil monitoring and integration with behavior was developed from scratch, with no examples from the literature to follow. Despite the small N, the results have quite large effect sizes and high statistical significance, supporting that the N is sufficient to test our hypotheses.

6) It would be useful validate that the measure of bias in Figure 1 is not sensitive to the number of trials in each loudness condition, as this will clearly differ across pupil bins. We are not sure about the total number of trials in some of the loudness conditions, but we worry a bit that if there are very few trials in a given loudness condition, the measure could be misleading.

Thank you for this point. We now write: “Since sensitivity increased with tone loudness (Figure 1—figure supplement 1D), there were more go-trials for louder tones (mean number of go-trials across subjects from least to most loud tones): 68, 73, 97, 122, 136, 166 and 45, 69, 89, 215 go-trials, for mice and human subjects respectively. To ensure an equal number of tones of each loudness in each pupil bin, we binned trials by pupil response within a given loudness condition.”

Thus, we don’t think the bias measure in Figure 1 is misleading. Computing SDT criterion per loudness condition, and then averaging across, gave very similar results (Figure 1—figure supplement 1H), and (ii) that loudness / difficulty was not a factor in any of the yes/no data sets, and we observed the same behavioral correlates of phasic pupil responses.

7) It would be useful to flesh out in a bit more detail why the observation of non-linear vs. linear arousal state modulation would indicate distinct functional roles for tonic vs. phasic arousal respectively. First, a linear relationship could just represent the increasing part of the inverted u-shape and secondly the original cited papers by Aston-Jones and Cohen, 2005; Yerkes and Dodson, 1908, used pupil size, while I believe the current work focuses on the speed of dilation. It would be important to clarify as to how these measures are related or how they may reflect activity of distinct neuromodulators or modulatory receptors as the authors pointed out in the final paragraph of subsection “In humans and mice, phasic arousal tracks a reduction of choice bias in an auditory detection task”.

We have added: “Second, pupil responses exhibited a negligible (go/no-go data sets) or weak (yes/no data sets) correlation to the preceding baseline diameter (Figure 1—figure supplement 1H, Figure 2—figure supplement 1D and Figure 3—figure supplement 1C); in all cases, these correlations are substantially smaller than those for conventional baseline-corrected pupil response amplitudes (de Gee et al., 2014; Gilzenrat et al., 2010).”

We have added: “It is tempting to speculate that these differences result from different neuromodulatory systems governing tonic and phasic arousal. Indeed, rapid dilations of the pupil (phasic arousal) are more tightly associated with phasic activity in noradrenergic axons, whereas slow changes in pupil (tonic arousal) are accompanied by sustained activity in cholinergic axons (Reimer et al., 2016).”

Finally, in the mouse go/no-go data set we were able to track the complete inverted-U relationship between performance and tonic arousal (tracked by pre-trial baseline pupil size) (McGinley et al., 2015). In the same data set, we here found a linear relationship between choice bias and phasic arousal (tracked by task-evoked speed of dilation). Together, this indicates that the linear relationship observed here does not represent the increasing part of the inverted-U.

Note that mice exhibited a wider variety of tonic arousal states during task performance than human subjects, ranging from almost falling asleep, to alert and focused performance, to running on the wheel. First, mice performed the go/no-go task sessions that each lasted around 60 minutes, while human subjects performed the various tasks in blocks of 10 minutes. Second, mice could also (and did) run on the wheel during the task, while human subjects were seated. Third, humans can likely sustain their attention for longer periods than mice. Thus, the tonic arousal state in human subjects was considerably more stable than that of mice. It is likely that for this reason we did not observe the same inverted-U relationship between tonic arousal and performance in humans as found for mice (data not shown).

8) The authors argue that experiment 3 shows reduced sampling from memory, but the idea that mnemonic decisions are based on samples from memory is still an early theory. Further, to our knowledge we don't know that the DDM does recover information about this memory sampling process (i.e. we do not know whether DDM drift biases reflect sampling from memory, sampling from the stimulus, or some other time-varying decisional process like response competition). It would be helpful to be more circumscribed in the interpretation of these results as reflecting a change in sampling from memory.

We fully agree that the idea that mnemonic decisions are based on sampling from memory is an assumption, which require further experimental validation. We have now toned down the corresponding conclusions accordingly (Discussion paragraph six).

9) What distribution was used to determine the length of mini-blocks? The Materials and methods say that miniblocks were never more than 7 trials, which suggests that the hazard rate for a signal trial was not constant across the miniblock.

We have presented the distribution that determined the length of mini-blocks in Figure 1—figure supplement 1A, left. The exact probabilities were 0, 0.228, 0.204, 0.18, 0.157, 0.133, 0.109, for trials 1 to 7 within a mini block, respectively. These were chosen to make the hazard rate more flat than a uniform distribution, but avoid long stretches of time without a target (which would occasionally occur if we used a flat hazard rate).

We have now plotted the hazard rate (Figure 1—figure supplement 1A, right) and agree it is not flat. It was partially flattened (compared to uniform probability across mini blocks). We now clarify this: “Although the hazard rate of signal probability increased within a mini block (Figure 1—figure supplement 1A, right), pupil responses did not consistently vary with trial number within a mini block (see Author response image 2).”

**Author response image 2. sa2fig2:** Task-evoked pupil response for mice (left) and human (right) subjects, separately for trial within a mini block. In the mouse data set, pupil responses did not systematically vary with trial number within a mini block: one-way repeated measures ANOVA F5,20 = 2.177, p = 0.098. For humans, they did: one-way repeated measures ANOVA F5,95 = 3.638, p = 0.005. Group average (N = 5; N = 20); error bars, s.e.m.

Also, the hazard rate for the reference trial is presumably zero, were these trials included in analysis?

Indeed, the hazard rate for the reference trial was zero. Therefore, these reference trials were excluded from the analyses. We now write: “In the go/no-go task, the first trial of each mini block (see Behavioral tasks) was excluded from the analyses, because this trial served as a mini block start cue, and never included the signal (pure sine wave).”